# Modeling Dynamic Neural Activity by Combining Naturalistic Video Stimuli and Stimulus-Independent Latent Factors

**Finn Schmidt**[1]
finn.schmidt@uni-goettingen.de

**Polina Turishcheva**[1]
turishcheva@cs.uni-goettingen.de

**Suhas Shrinivasan**[1]
shrinivasan@cs.uni-goettingen.de

**Fabian H. Sinz**[1,2]
sinz@uni-goettingen.de

[1] Institute of Computer Science and Campus Institute Data Science, University of Göttingen, Germany
[2] Lower Saxony Center for AI & Causal Methods in Medicine, Germany

## Abstract

The neural activity in the visual processing is influenced by both external stimuli and internal brain states. Ideally, a neural predictive model should account for both of them. Currently, there are no dynamic encoding models that explicitly model a latent state and the entire neuronal response distribution. We address this gap by proposing a probabilistic model that predicts the joint distribution of the neuronal responses from video stimuli and stimulus-independent latent factors. After training and testing our model on mouse V1 neuronal responses, we find that it outperforms video-only models in terms of log-likelihood and achieves improvements in likelihood and correlation when conditioned on responses from other neurons. Furthermore, we find that the learned latent factors strongly correlate with mouse behavior and that they exhibit patterns related to the neurons' position on the visual cortex, although the model was trained without behavior and cortical coordinates. Our findings demonstrate that unsupervised learning of latent factors from population responses can reveal biologically meaningful structure that bridges sensory processing and behavior, without requiring explicit behavioral annotations during training. **Code:** github.com/sinzlab/SchmidtEtAl2025_Dynamic_Latent_State

## 1 Introduction

Predicting the activity of sensory neurons is a major goal of computational neuroscience towards understanding the mechanisms of information encoding in the brain. In particular, accurately predicting neural responses in the primary visual cortex (V1) in response to a given stimulus could provide deeper insights into how the brain processes visual information. However, this task remains challenging because neural activity in the visual cortex exhibits variability not only across different visual stimuli but also across repeated presentations of the same stimulus [38, 42]. This variability arises from the influence of numerous unobservable factors that the visual cortex integrates into its processing. For example, the activity in the visual cortex is influenced by factors such as behavioral tasks [13, 22], attention [8, 12, 31], and general brain states correlated with behavior [15, 32, 35, 40]. Recording all of these additional variables is not feasible, especially because most of them are unknown [40]. Thus, understanding the sensory processing of the visual cortex requires models that capture both stimulus-dependent and shared stimulus-conditioned variability [5]. One way to do

39th Conference on Neural Information Processing Systems (NeurIPS 2025).

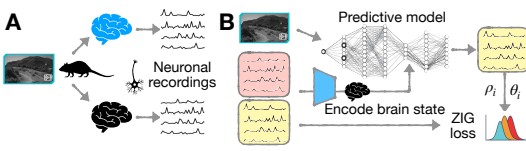

Figure 1: **A**. Problem statement: an animal watches the same video, but different brain states lead to different neuronal responses. **B**. Proposed solution: infer the brain state with a probabilistic latent-variable model that predicts marginal Zero-Inflated-Gamma (ZIG) distributions for each neuron.

it is to model internal fluctuations using latent variables that capture a shared state across neurons, explaining the variability in responses to the same stimulus. To this end, such models require a probabilistic framework to effectively incorporate and infer these latent variables.

The majority of video encoding models for neuronal activity in visual cortex focuses on models that predict neural activity conditioned solely on the stimulus [39, 43, 47] or on mapping neural activity to meaningful low-dimensional latent spaces, independent or only conditioning on the underlying task or stimulus [20, 24, 36, 41]. To the best of our knowledge, only a few models exist that capture the correlated variability of large neuronal populations and the visual stimuli as input [5]. None of them are designed for dynamic video-based stimuli.

In this paper, we address this gap with the following contributions:

- We propose a probabilistic neuronal predictive model that accounts for both dynamic visual stimuli and latent brain state. The brain state is extracted from a subset of neuronal activity (Fig. 1).
- Our model surpasses comparable models without latent in predicting neuron response distributions in terms of log-likelihood.
- We show that the model's latent variables are meaningful, as they are strongly correlated with mouse behavior (as expected from experimental work [32, 35, 40]) and exhibit topographic patterns along the cortical surface despite no exposure to behavioral data or cortical position data.

## 2   Related work

Existing data-driven neural response prediction methods fall into three categories:

❶ deterministic models that predict neural activity from visual stimuli but do not account for response uncertainty and variability [e.g. 3, 23, 39, 43, 46, 47],

❷ probabilistic models focused on deriving a latent state from neural responses, sometimes conditioning on the external sensory inputs [20, 36, 41, 55],

❸ models, using both sensory input and neuronal activity to predict other neurons' responses [5, 25].

App. A summarizes related works in terms of inputs, outputs, and presence of latents.

❶ **Deterministic models for visual cortex**   Recent advances in deep learning, particularly convolutional neural networks (CNNs) trained on image recognition tasks [10, 11, 33, 52], have significantly improved predictive models. However, while task-driven models work well for responses from macaque visual cortex, they do not necessarily work as well for mouse visual cortex [9]. Since our focus is on recordings from the mouse visual cortex, we focus on data-driven models, trained end-to-end on neuronal activity in the following. Early work focused on developing specialized stimulus feature representations [called *cores*, 2, 16] or readout architectures [27, 29] that map the output of the core to neuronal representations. Sinz et al. [39] extended the core to video stimuli, adding a gated recurrent unit, and incorporating behavioral data such as pupil dilation and running speed. The inclusion of these behavioral variables allows the model to capture aspects of latent brain state and their interaction with neuron selectivity. For instance, Franke et al. [17] showed that behavioral activity modulates stimulus selectivity in mouse visual cortex when processing colored natural scenes. Höfling et al. [23], Vystrčilová et al. [46] further improved core architectures with factorized 3D convolutions for dynamic stimuli, achieving faster training and better performance than GRU-based models. In parallel, Wang et al. [47] introduced a foundational model for mouse visual cortex with a preprocessing module that adjusts for pupil position and integrates recurrent components into the core. Recently, transformer-based models [3, 28] have shown promising results.

While these models reproduce biological phenomena and achieve state-of-the-art performance, they all train point estimators, i.e., deterministic networks that predict a single value for each trial. The lack of a proper probabilistic model prevents them from computing log-likelihoods and limits their ability to model the stochastic distribution of neuronal activity. Wu et al. [51] proposes a variational

model for the network weights to model neurons' distributions, but this model only takes visual stimuli as input and does not model an independent latent state.

❷ **Probabilistic models focused on a latent state** Previous work has made significant progress in reducing high-dimensional neuronal recordings to smooth low-dimensional manifolds, often interpreted as latent states. However, these models do take the animal's sensory input into account, which is one strong driving force of neuronal activity, in particular in sensory areas. Yu et al. [54] introduced Gaussian Process Factor Analysis to map neural activity over time into a low-dimensional space, assuming a Gaussian distribution for neural activity given the latent variable at each time point. Jensen et al. [24] scaled this approach using variational inference. Gokcen et al. [20] extended further by incorporating time delays to model interactions between neural populations, revealing distinct latent dimensions for shared and region-specific neural dynamics. Building on this, Gokcen et al. [21] also introduced a shared latent space across sessions. However, these models assume a simple linear-Gaussian relationship between latent variables and neural activity.

To extend these approaches to nonlinear modeling, other works [36, 41, 56, 57] use variational autoencoders. The encoder learns the approximate posterior $q(z|y, a)$ for latent variables $z$ based on neural responses $y$ and auxiliary data $a$, while the decoder reconstructs neural data from $z$. Sussillo et al. [41] uses a generative recurrent network to model temporal dependencies in $z$ for neuronal spikes. Zhu et al. [57] extended this work for calcium recordings using zero-inflated gamma distribution as a loss. In contrast, Schneider et al. [36] trains with contrastive loss, integrating behavior or task information to produce similar embeddings for comparable neural activity. Similarly, Zhou and Wei [56] models dependencies between task labels $u$, latents $z$, and observations $x$ with a two-stage non-contrastive approach. However, those methods are focused on encoding neuron responses into meaningful latent embeddings and do not predict neuron responses based on external stimuli inputs like images or videos.

❸ **Models using sensory and neuronal input to predict other neurons' activity** Recent work on recurrent state space models (RSSMs) used latent states for neuronal response modeling. For instance, Glaser et al. [19] uses observed neural activity and exogenous factors to derive a latent state. However, their model is limited to discrete states. Zoltowski et al. [58] are able to predict both discrete and continuous random states. However, their model is limited to a decision-making task. Apart from RSSMs, Kim et al. [25] uses a flow model, taking both task-relevant inputs and some neurons to predict responses of the other neurons, but focuses on the auditory cortex, where both task and neuronal inputs are time series. To our knowledge, Bashiri et al. [5] is the only work combining visual input with a latent state. It extends the core-readout framework by adding flow-modeled, stimulus-conditioned variability, including noise correlations, to the mean activity predicted by the core-readout model. However, the model of Bashiri et al. [5] is limited to static stimuli. Since their approach requires a covariance matrix between the responses of all neurons to an image, it scales quadratically with the number of neurons. This makes it computationally too expensive for temporal dependencies, since it would scale quadratically in the number of neurons and timepoints.

## 3 Proposed latent model

In this work, we modify a well-established deterministic core-readout framework — consisting of a factorized 3D convolutional core [46] and a Gaussian readout [29]— to a probabilistic latent state architecture (details in App. C). Specifically, we add an encoder that takes a subset of neurons as input, reduces their dimensionality, and derives a latent variable. This latent state is then combined with the transformed visual input to predict the activity of other neurons from the same session. This model is equivalent to a latent variable model that predicts neuronal activity from two independent factors i) latent state and ii) the input video. In contrast to Bashiri et al. [5], our model is video-based and predicts time series of neuronal response and the corresponding marginal distribution for each neuron and time point. We designed the model to predict a neuron's activity conditioned on any subset of neurons from the same experiment. In addition, it can explicitly infer the latent state, allowing us to analyze its relation to other external factors such as the behavior of the animal.

**Model architecture** Our model predicts time-varying neuronal responses $\mathbf{y} \in \mathbb{R}^{N \times T}$ to a video stimulus $\mathbf{x} \in \mathbb{R}^{W \times H \times T}$, where $N$ is the number of neurons, $T$ the number of time points, and $W$ and $H$ are the frames' width and height. The prediction also depends on a dynamic, stimulus-independent latent factor $\mathbf{z} \in \mathbb{R}^{k \times T}$ with latent dimension $k \ll N$ (Fig 2). To obtain a probabilistically grounded

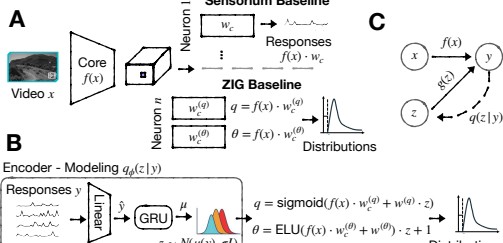

Figure 2: **A**. A 3D convolutional neural network-based core extracts features of the video input $\mathbf{x}$. The read-out learns the spatial position (purple dot). The deterministic Poisson baseline predicts the mean response via a per-neuron affine function from the learned feature vector $\mathbf{w}^c$. For the ZIG baseline, we double the dimension of $\mathbf{w}^c$ to $\mathbf{w}^{(q,\theta)}$ to predict response-distribution parameters $q, \theta$ of a ZIG distribution.

**B**. For the latent model we keep the core-read-out architecture $\mathbf{w}^{(q,\theta)}$ and additionally add the latent variable $\mathbf{z}$ to predict $q, \theta$. The encoder (approximate posterior) first reduces the dimensionality of masked neuronal responses $\mathbf{y}$. Afterwards, a GRU computes the posterior $q(\mathbf{z}|\mathbf{y}) \sim \mathcal{N}(\mu(\mathbf{y}), \sigma\mathbf{I})$, from which we sample the latent state when conditioning on neurons. For the marginal distribution $p(\mathbf{y} \mid \mathbf{x})$ we sample the latents from the prior $\mathcal{N}(0, \mathbf{I})$ and integrate out $\mathbf{z}$ via Monte-Carlo sampling. **C**. Directed graphical model of video, latent, and response variables; the dashed line indicates the variational approximation.

latent state, we combine ideas from Bashiri et al. [5] and Zhu et al. [57] by modeling neuronal responses conditioned on stimulus and latent factors using a Zero-Inflated Gamma (ZIG) distribution, which provides a more realistic assumption for calcium imaging data compared to a widespread Poisson assumption. In calcium imaging of neurons, the deconvolved fluorescence signals often contain many zero or near-zero values when neurons are inactive, together with positively skewed values when activity occurs. The ZIG distribution naturally captures this behavior by combining a point mass at zero (to model silent periods) with a gamma distribution that represents the continuous, positive-valued calcium activity during neural firing. ZIG is a mixture of a uniform distribution for zero responses and a shifted Gamma distribution for nonzero neuronal responses [49]:

$$p_{\text{ZIG}}(\mathbf{y}|\mathbf{x}, \mathbf{z}; \psi) = \prod_{y_{it} \leq \rho}^{N,T} \left( \left(1 - q_{it}(\mathbf{x}, \mathbf{z}; \psi)\right) \cdot \frac{1}{\rho} \right) \cdot \prod_{y_{it} > \rho} \left( q_{it}(\mathbf{x}, \mathbf{z}; \psi) \cdot f_\Gamma \left( y_{it}; \rho, \kappa_i, \theta_{it}(\mathbf{x}, \mathbf{z}; \psi) \right) \right)$$

where $\rho$ is simultaneously the width of the uniform part and the shift of the Gamma distribution, $\kappa_i$ the shape parameter per neuron $i$, $q_{it}$ the nonzero response probability, and $\theta_{it}$ the scale parameter for each neuron at time-step $t$. $f_\Gamma(\cdot; \rho, \kappa, \theta)$ is the probability density function of a Gamma distribution shifted by $\rho$. Our latent model predicts the distribution parameters $\boldsymbol{\theta}$ and $\mathbf{q}$ based on a given video-stimulus, the encoded latent factors, and the model's parameters $\psi$:

$$q_{it}(\mathbf{x}, \mathbf{z}, \psi) = \text{sigmoid}\left( f_{it}^{(q)}(\mathbf{x}; \psi) + \mathbf{w}_i^{(q)} \cdot \mathbf{z}_t \right), \theta_{it}(\mathbf{x}, \mathbf{z}, \psi) = \text{ELU}\left( f_{it}^{(\theta)}(\mathbf{x}; \psi) + \mathbf{w}_i^{(\theta)} \cdot \mathbf{z}_t \right) + 1. \tag{1}$$

$\kappa_i$ is fitted independent of the stimulus once per neuron on the train set via moment matching to avoid un-identifiability conditions during model training (i.e. mean of the Gamma distribution $\kappa \cdot \theta$ can be realized by infinitely many combinations of $\kappa$ and $\theta$). For each neuron and time point, we use the mean of the predicted ZIG distribution as the response prediction.

We model $f_{it}^{(q)}$ and $f_{it}^{(\theta)}$ using a typical core-readout architecture (Fig 2A) where the core extracts features of the video-input, and the readout maps the relevant features from the core-output to the individual neurons. While the core has the same architecture as a model without a latent state, we doubled the feature vectors of the classical Gaussian readout to predict both $q$ and $\theta$. We additionally tested to decode the latent variables replacing $\mathbf{z}_t$ with smoothed latent variables $g(\mathbf{z}_t)$ in Eq. (1).However, this did not improve the models' performance (App. D). The stimulus-independent latent factors are modeled with an isotropic Gaussian prior across time and factors $p(\mathbf{z}) = \mathcal{N}(0, \mathbf{I})$. We model an approximate posterior (Fig. 2B) as a Gaussian with the mean as a function of the responses $\mathbf{y}$, the encoder parameters $\phi$, and an independent variance: $q(\mathbf{z}|\mathbf{y}; \phi) = \mathcal{N}(\mu(\mathbf{y}; \phi), \sigma^2\mathbf{I})$.

**Training** We fit the model by maximizing $p_{\text{ZIG}}(\mathbf{y}|\mathbf{x})$ via its evidence lower bound (ELBO) via variational inference [7, 26]:

$$\log p_{\text{ZIG}}(\mathbf{y}|\mathbf{x}) \geq \langle \log p(\mathbf{y}|\mathbf{z}, \mathbf{x}) \rangle_{\mathbf{z} \sim q_\phi(\mathbf{z}|\mathbf{y})} + D_{\text{KL}} \left[ q_\phi(\mathbf{z}|\mathbf{y}) : p(\mathbf{z}) \right], \tag{2}$$

where $\langle \cdot \rangle$ represents expected value and $D_{\text{KL}}$ the Kullback-Leibler divergence. The likelihood term $\langle \log p(\mathbf{y}|\mathbf{z}, \mathbf{x}) \rangle_{\mathbf{z} \sim q_\phi(\mathbf{z}|\mathbf{y})}$ of the ELBO is computed via Monte-Carlo sampling by drawing samples

$\mathbf{z}$ from the approximate posterior $q_\phi(\mathbf{z}|\mathbf{y})$. We found that 150 samples are optimal (App. B). Since $q_\phi(\mathbf{z}|\mathbf{y}), p(\mathbf{z})$ are Gaussians, we computed the Kullback-Leibler divergence analytically. In all of the experiments, the latent model was initialized with a pretrained video-only ZIG model, unless stated otherwise.

**Marginalization**   To compare the latent model with the baselines, which do not take neuronal activity as input, we sample the latent from the prior and marginalize $\mathbf{z}$ via Monte-Carlo sampling to compute the response likelihoods given only the video $p(\mathbf{y}|\mathbf{x})$:

$$p(\mathbf{y}|\mathbf{x}) = \int p(\mathbf{y}|\mathbf{x}, \mathbf{z})p(\mathbf{z})\, d\mathbf{z} = \frac{1}{L} \sum_l^L p(\mathbf{y}|\mathbf{x}, \mathbf{z}^{(l)}). \tag{3}$$

App. B shows that the log-likelihood stabilizes and plateaus after approximately 1000 samples.

# 4   Experiments

**Dataset**   We trained and evaluated our models on the data from the five mice of the SENSORIUM competition [43]. The SENSORIUM IDs of the mice are in App. F. Responses are 2-photon calcium traces from $\sim 40,000$ excitatory neurons of five mice in layers 2–5 of the primary visual cortex. They were recorded at 8 Hz while the head-fixed mice viewed naturalistic gray-scale videos at 30 Hz. The signals were synchronized and upsampled to 30 Hz. During training, we used video input with shape $(W, H, T) = (64, 36, 80)$, while for evaluation, we used the whole length $T$ for each video. Behavioral variables—locomotion speed, pupil dilation, and center position—were also recorded and resampled to 30 Hz.

**Baseline models**   To compare with the standard non-probabilistic model, we use the SENSORIUM [43] baseline model with a Poisson loss. To disentangle the impact of the ZIG loss and the latent modeling, we use a video-only ZIG-model (without latent) as a second baseline. All models share the same hyperparameters wherever possible (Tab. 8). Since we aim to use correlations between latent variables and behavioral variables [5, 40] as external validation for the model (see below), we excluded behavioral data during training for all models – in contrast to previous work [39, 47].

**Evaluation**   Building upon prior work [39, 43, 50], we use single-trial-correlation averaged across neurons to measure model performance. For each neuron $i$, we compute the correlation between the neuron response predictions $\hat{y}_{jti}$ and the actual responses $y_{jti}$ over all time points $t$ of all videos $j$ in the evaluation data. We evaluate the model in two scenarios. First, we want to compare the model with the baselines, which do not have any additional neuronal input. Hence, in the "Non-Conditioned" scenario, we sample latents $\mathbf{z}^{(l)}$ from the prior and use the mean of the response distribution as a predictor

$$\hat{\mathbf{y}}_{ji} = \int \mathbf{y}_i \cdot p(\mathbf{y}_i|\mathbf{z}, \mathbf{x}_j)p(\mathbf{z})\mathrm{d}\mathbf{z} = \langle \mathbf{y}_i \rangle_{\mathbf{y}_i \sim p(\mathbf{y}_i|\mathbf{z}, \mathbf{x}_j)p(\mathbf{z})}.$$

Here, $\hat{\mathbf{y}}_{ij}$ is the predicted time series of the response of neuron $i$ to video $j$.

Then we want to check if adding responses as input helps. So, in the second —"Conditioned"— scenario, we derive the latent state from a subset of the population of neurons $\hat{\mathbf{y}}_{ji} = \langle \mathbf{y}_i \rangle_{\mathbf{y}_i \sim p(\mathbf{y}_i|\mathbf{z}, \mathbf{x}_j)p(\mathbf{z}|\mathbf{y}_\mathcal{I})}$ where $\mathcal{I}$ is an index set. Algorithmically, the only difference between the scenarios is how we sample $\mathbf{z}$.

As we model full response distributions, we also compute the log-likelihood to evaluate how well those distributions are predicted. For our latent model, the latent variables $\mathbf{z}$ are marginalized out via Monte-Carlo sampling (Eq. (3); for experiment details see App. E.

**Non-Conditioned evaluation: Latent model improves response distribution modeling**   The latent model outperforms the video-only ZIG model in terms of log-likelihood (Tab. 1), demonstrating the improved capability of the latent model for capturing the full response distributions. Since the neuronal responses are continuous and the Poisson distribution is for discrete values only, evaluating the log-likelihood of the Poisson model is not meaningful.

Adding a latent to our ZIG model does not hurt predictive performance and yields comparable correlation. The ZIG and latent models have about one percentage point lower correlation compared to the Poisson baseline model (Tab. 1). This happens because the ZIG distribution is not an

exponential family, which leads to a trade-off between modeling full distributions (likelihood) and conditional means (correlation) [30]. We test this claim by training the means of the predicted ZIG distribution with Poisson loss ('Poisson loss ZIG'), which leads to a comparable performance with the Poisson baseline model. Details are in the App. G.

To test if there is a better distributional fit than the Gamma distribution for the positive responses, we replaced it with a more flexible normalizing-flow–based distribution while keeping everything else fixed (latent-state and stimulus dependence). However, adding flow layers did not improve the performance of our ZIG model, suggesting that the ZIG distribution already provides an adequate fit to the neuronal response distributions (App. H).

Table 1: Predictive performance of models. Log-likelihood is computed in Bits per Neuron and Time. Standard error of the mean (SEM) is reported across three models with different initializations.

|  | Poisson Baseline | Poisson Loss ZIG | Video-only ZIG | Latent ZIG |
|---|---|---|---|---|
| Pearson Correlation ↑ | **0.195** ±0.003 | **0.194** ±0.003 | 0.183 ±0.003 | 0.182 ±0.004 |
| Log-Likelihood ↑ | – | – | -0.98 ±0.04 | **-0.30** ±0.05 |

Finally, we tested whether our latent model also improves performance when combined with a different core—specifically, a Vision Transformer (ViT) core as in [28]. The quantitative trends are the same: the ZIG model with the ViT core performs slightly worse than the Poisson baseline due to the optimization trade-off, whereas the latent model improves performance beyond the Poisson baseline (see App. I for details).

**Distributional modeling helps on out-of-distribution stimuli**    Point-estimate models, like our Poisson baseline model, empirically struggle on OOD stimulus distributions such as gratings or moving dots (Tab. 2 'Poisson'[1]) when response distributions shift with stimuli. Since ZIG and latent models learn per-neuron distributions, we hypothesized that they might generalize to OOD stimuli.

To assess generalization, we evaluated our models on the SENSO-RIUM bonus track, which provides neural recordings collected in response to out-of-distribution (OOD) video stimuli-such as pink-noise clips, whose spatiotemporal statistics significantly diverge from the statistics of naturalistic videos used during training. Learning per-neuron distributions indeed improves OOD performance (Tab. 2, 'ZIG'). The latent model performs similarly when sampling from the prior (Tab. 2, 'Non-Conditioned'). Conditioning on half the neurons further boosts correlations (Tab. 2,'Con-

Table 2: Correlations on half of the neurons for out-of-distribution (OOD) stimuli, averaged across three OODs per mouse: (1) Poisson baseline, (2) ZIG model, (3) Latent ZIG without conditioning, (4) Latent ZIG conditioned on half of the neurons. SEM is across mice.

|  | Poisson | ZIG | Non Conditioned | Conditioned |
|---|---|---|---|---|
| Mouse 1 | 0.058 | 0.091 | 0.091 | 0.156 |
| Mouse 2 | 0.097 | 0.128 | 0.128 | 0.174 |
| Mouse 3 | 0.110 | 0.134 | 0.133 | 0.181 |
| Mouse 4 | 0.084 | 0.118 | 0.115 | 0.185 |
| Mouse 5 | 0.064 | 0.091 | 0.087 | 0.153 |
| Average | 0.08±0.009 | **0.11**±0.008 | **0.11**±0.007 | **0.17**±0.009 |

ditioned'), showing that encoding neuron responses also generalizes to OOD.

**Increasing the latent dimension improves performance in the 'Conditioned' scenario but could lead to overfitting**    In the 'Conditioned' scenario, the dimensionality $k$ of the latent variable is the information bottleneck: a bigger latent dimension helps to pass more information from the neuronal input. However, it requires approximating a higher-dimensional distribution, which might be vulnerable to overfitting. To explore the optimal latent dimension, we assessed the predictive correlation of models with (1) different latent dimensions $k$ and (2) varying portions of responses $\mathbf{y}_{\mathcal{I}}$ as input. Throughout, we report the correlation on a quarter of the neuron population ($n \approx 2000$) per mouse, which was never used as input, i.e., where not part of the index set $\mathcal{I}$. Given half of the neurons responses as input in the 'Conditioned' scenario (Fig. 3 A), our latent model achieves a performance of up to 0.27 ($\pm0.003$), a boost of 0.09 compared to the video only prediction performance of 0.18. However, in the 'Non-Conditioned' scenario, increasing the latent dimension actually decreases the correlation from 0.18 of the latent model to 0.16 (Fig. 3 A). Beyond $k = 80$ we observe no significant changes. A reason for the decrease in the 'Non-Conditioned' scenario could be that increasing

---

[1]The Poisson baseline numbers are lower than in [43] as we do not use behavior data

the latent dimension enhances the latent model's encoding capabilities, while the video-processing component remains unchanged. As a result, the model may rely too much on the encoded responses in the latent variables, which might hurt the correlation when sampling from the prior.

We further explore the effect of conditioning on different portions of the neuron population on the predictive performance of a latent model with a low latent dimension ($k = 5$) and a high latent dimension ($k = 100$).

For the low-dimensional latent model, the performance levels off when conditioned on 30% of the neuron population, reaching a correlation of about 0.23 ($\pm 0.004$). In contrast, the high-dimensional latent model keeps improving with more neurons given to the encoder, finally reaching a correlation of 0.28 ($\pm 0.004$) when 75% of neuron responses are given (Fig. 3 B).

In App. J, we further tested whether our baseline models achieve comparable performance if we include behavioral variables as a replacement for the latent state. However, the respective models do not exceed a correlation of 0.20. This indicates that our latent state encodes a high-dimensional brain state that is not captured by few behavioral dimensions, in line with previous work [40].

**Increasing latent dimension improves log-likelihood in 'Non-Conditioned' scenario**     Because the correlation decreases with the latent dimensionality in the 'Non-Conditioned' scenario, one might expect a decline in log-likelihood performance as well, while it is actually growing with a higher latent dimension (Fig. 3 C). To analyze this apparent discrepancy, we checked how the variance of the latent factors affects the log-likelihood. Since the average norm of the latent feature vectors $\mathbf{w}_i^{(q,\theta)}$ increases with the latent dimension, the variance of distribution $\mathbf{w}_i \cdot \mathbf{z}_t \sim \mathcal{N}(0, \|\mathbf{w}_i\|_2^2)$ increases as well (Fig. 3 C).

Eq. (1) show that if $\mathbf{w}_i \cdot \mathbf{z}_t$ has an increased variance, we sample the distribution parameters $q, \theta$ from a broader range. Hence, a high-dimensional model is more likely to put probability mass on more extreme response values and does not concentrate all the probability mass around the mean. Examples can be seen in App. K. This improves the log-likelihood $\log p(\mathbf{y}|\mathbf{x}) \approx \log\left(\sum_l p(\mathbf{y}|\mathbf{x}, \mathbf{z}^{(1)})\right) - \log(L)$, since its more likely to sample a probability density function $p(\cdot|\mathbf{x}, \mathbf{z}^{(1)})$ with a positive value at $\mathbf{y}$ even for extreme response values $\mathbf{y}$. Thus, a high-dimensional model predicts more realistic response distributions by better accounting for potential outliers.

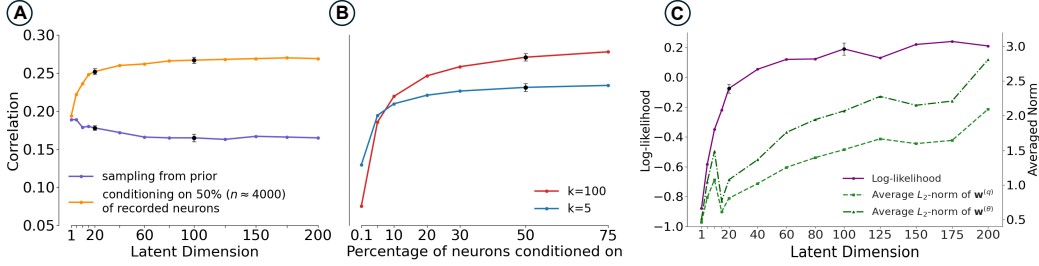

Figure 3: **A** Average prediction–response correlation across models with varying latent dimensions. **B** Average conditioned correlation for low- vs. high-dimensional latent models, if different portions of the neuron population are given. **C** Log-likelihood vs. average norm of latent feature vectors $\mathbf{w}_i^{(q)}, \mathbf{w}_i^{(\theta)}$ across latent dimensions. Error bars: SEM over 3 seeds; some points omitted to save training cost.

**Latents primarily reflect a stimulus-independent brain-state**     If we sample from the prior, the latent is independent from the video input a priori. However, if we condition the latents on neuron responses, i.e., sampling the latents from the approximate posterior $q(\mathbf{z}|\mathbf{y})$, $q(\mathbf{z}|\mathbf{y})$ can still carry stimulus information due to the collider structure $\mathbf{x} \rightarrow \mathbf{y} \leftarrow \mathbf{z}$ in the graphical model (Figure 2C).

To test whether latents sampled from $q(\mathbf{z}|\mathbf{y})$ reflect stimulus-independent brain states rather than residual stimulus content, we ran two control experiments. In the first experiment, we want to ensure that we cannot predict the latent state from the video statistics, therefore, we compute the local contrast $c_{it}$ for each time point $t$ and each neuron $i$ measured via the pixel variance in the receptive field of the neuron for the given video frame.

The receptive field is determined by the maximal receptive field of the convolutional core—as determined by the kernel sizes—at the readout location of the neuron.

We performed a linear regression $\text{argmin}_{W,b} \sum_t ||W\mathbf{c}_t + \mathbf{b} - \mathbf{z}_t||_2$ and computed the $R^2$ values of the actual latents $\mathbf{z}_t$ and the predicted values $\hat{W}\mathbf{c}_t + \hat{\mathbf{b}}$ on the test set. We performed a 5-fold cross-validation on the validation data to compute the average $R^2$ values. For 4 of the 5 mice, the video contrast does not explain the variance of the latents. However, for one mouse we get values up to 0.27 ($\pm 0.01$) indicating that the latents sometimes do encode residual stimulus information (Tab. 3 Col. 1).

In the second experiment, we use repeated stimuli to validate that the model encodes brain state-related and not visual-input-related information. We split the neuronal population into two halves $\mathbf{y} = (\mathbf{y}_{\text{one\_half}}, \mathbf{y}_{\text{other\_half}})$, shuffled the matching of the halves across trials that presented the same stimulus, and used the shuffled pairs to predict $\mathbf{y}_{\text{one\_half}}$ via $p(\mathbf{y}_{\text{one\_half}}^{(i)} \mid \mathbf{x}, \mathbf{y}_{\text{other\_half}}^{\pi(i)})$, where $i$ and $\pi(i)$ denote the index of the original and shuffled trial. This approach ensures that $\mathbf{y}_{\text{other\_half}}$ corresponds to the same video but a different brain state. If $\mathbf{z}$

Table 3: (1) Contrast $R^2$; (2) Conditioned correlation; (3) Correlation for mixed repeats; (4) Correlation int the Non-Conditioned scenario. SEM averaged over neurons and videos.

| | Contrast $(R^2)\downarrow$ | Corr. same rep.↑ | Corr. mixed rep.↓ | Corr. Non-Conditioned↑ |
|---|---|---|---|---|
| Mouse 1 | 0.05±0.03 | 0.17±0.02 | 0.12±0.01 | 0.12±0.02 |
| Mouse 2 | −0.19±0.20 | 0.19±0.02 | 0.16±0.02 | 0.15±0.01 |
| Mouse 3 | 0.27±0.01 | 0.20±0.02 | 0.17±0.02 | 0.16±0.01 |
| Mouse 4 | −0.03±0.04 | 0.24±0.03 | 0.21±0.02 | 0.18±0.01 |
| Mouse 5 | 0.01±0.04 | 0.21±0.02 | 0.15±0.02 | 0.14±0.01 |
| Average | 0.02±0.07 | 0.20±0.01 | 0.16±0.01 | 0.15±0.009 |

primarily captures video responses, the performance should not decline significantly. Conversely, if $\mathbf{z}$ mostly reflects brain state, performance should drop to the level without conditioning—or worse. Tab. 3 Col. 2,3 shows that correlation drops for all mice when conditioning on mixed repeats versus same-repeat responses, indicating that the latents encode a stimulus-independent brain state.

**Sanity check: Latent variables strongly correlate with behavior** It is widely known that latent brain state is strongly correlated with behaviour [5, 32, 40], therefore, we use behaviour to test whether the latent variables sampled from our approximate posterior capture relevant internal states of the brain. Specifically, we computed a canonical correlation analysis (CCA), which finds the linear combination of the latent variables $\mathbf{z}^{(1)}, \ldots, \mathbf{z}^{(k)}$ with maximal correlation to a chosen behavioral variable like pupil dilation or treadmill speed.

Although our model has not seen any behavioral data during training, the learned latents show strong correlations with behavioral data (Fig. 4), suggesting that our latents indeed encode meaningful brain-state related information. Analysis details are in App. L.

To assess how meaningful these correlations are, we compare against a CEBRA baseline [36] trained on the same data, with hyperparameters tuned to maximize behavioral correlation and without visual-stimulus input or held-out neurons (details in App. L). While the CEBRA model attains slightly higher correlation with pupil dilation, our latent model achieves much higher and more stable correlation with treadmill speed, yielding overall comparable performance (Table 4).

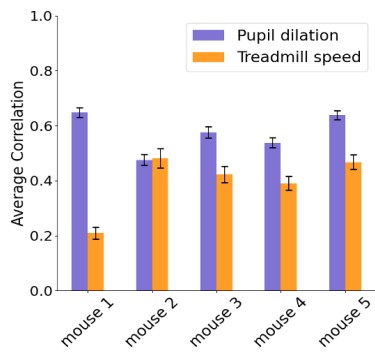

Figure 4: Canonical correlation of behavior and latent for each mouse. The CCA analysis was done with 5-fold cross-validation in an 80/20 split. The error bars indicate the standard error of the mean of cross-validation.

Table 4: Average canonical correlation across all mice for our latent model (ZIG latent) and a CEBRA model trained on the same data. SEM is computed over the different validation correlations in the cross-validation.

| | CEBRA | ZIG latent |
|---|---|---|
| Pupil dilation ↑ | **0.66** (0.004) | 0.60 (0.01) |
| Treadmill speed ↑ | 0.27 (0.006) | **0.42** (0.02) |

**Feature vectors of a latent model exhibit topographic organization** Bashiri et al. [5] analyzed the relation between latent state and position in the visual space or cortex. This inspired us to explore whether our latent space relates to the cortical positions of neurons as well. The latent space model has learned two feature matrices, $\mathbf{w}^{(\theta)}$ and $\mathbf{w}^{(q)}$, where the columns $\mathbf{w}_i^{(\theta)}, \mathbf{w}_i^{(q)}$ are the feature vectors

for each neuron $i$. These vectors serve as neuron-specific weights that project the shared latent state onto that neuron's response-distribution parameters (see Eq. (1)).

First, we visualized these vectors and noticed a spatial organization on the cortical surface. Specifically, we computed the most relevant direction in the weight spaces by a singular value decomposition (SVD) on the feature matrices, extracting the singular vectors $u_1, u_2, \cdots \in \mathbb{R}^k$ that explain most of the variance in the weight vectors. Subsequently, we project each weight vector onto the first three SVs via $u_j \cdot \mathbf{w}_i^{(q)}$ and $u_j \cdot \mathbf{w}_i^{(\theta)}$ for $j = 1, 2, 3$ and each neuron $i$.

Then we plotted the neurons using their cortical xy positions and using the SV values as colors (Fig. 5 Columns 1,2). The visualization for $\mathbf{w}^{(q)}$ is in Fig. 5, similar visualizations for $\mathbf{w}^{(\theta)}$ are provided in App. M.

Although our model was not explicitly provided with cortical position data during training, it still learned patterns that suggest a spatial organization by cortical position (Fig. 5, Col. 1-3). These patterns appear independent of the neurons' depth in the cortex (Fig. 5, Col. 1 and 2). Thus, a neuron's position seems to contain relevant information for predicting its responses. Beyond the third singular dimension, we did not find more spatially organized patterns (App. M).

**Sharing parameters via cortical coordinates reduces model size with minimal accuracy loss.** The previous experiment suggests that cortical positions are important for latents, hence, we could use them to learn the latent feature vectors and decrease the model size.

Instead of storing a separate pair of feature vectors $\mathbf{w}_i^{(q,\theta)}$ for every neuron, we let two small 2-layer MLPs map the neuron's cortical position $(p_x, p_y, p_z)$ to $\mathbf{w}_i^{(q,\theta)}$. This substitution removes thousands of free parameters, as we remove the big feature matrices, each of size $k \times N$ (latent dimension $\times$ number of neurons), as model parameters for each of the five mice. The decrease in log-likelihood and conditioned correlation is only marginal (Tab. 5). This confirms that neighboring neurons naturally have similar latent feature vectors. The topographic maps of coordinate-based $\mathbf{W}^{(q,\theta)}$ (Fig. 5, column 4) closely match those from the response-based (freely learned) latent model, indicating that these structures are

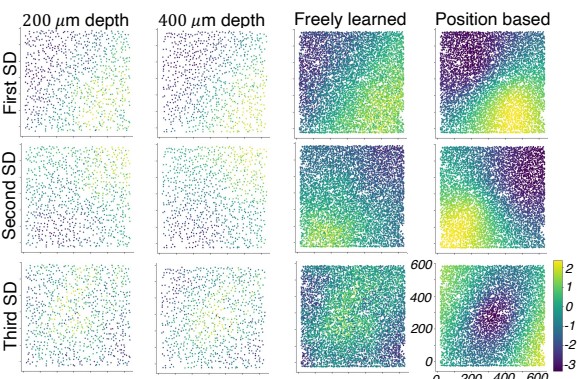

Figure 5: Color gradient maps of the latent feature vectors $\mathbf{w}_i^{(q)}$ along the first three Singular Dimensions for mouse 9. All recorded neurons are located within a $600\mu m \times 600\mu m$ square in the cortex. Their depth differs at most $200 \ \mu m$. The first three columns display the color maps for a model with freely learned feature vectors, trained without any knowledge of cortical positions. Columns 1-2 use only neurons from a specific depth for both SVD and visualization; Column 3 includes all neurons for both. Column 4 shows a model predicting latent features from cortical positions. Rows show the top three singular dimensions in descending order.

Table 5: Comparison of log-likelihood (in bits per Neuron and Time) and conditioned correlation of models with differently learned latent feature vectors $\mathbf{w}^{(\theta)}, \mathbf{w}^{(q)}$. The feature vectors are learned as independent model parameters per neuron (freely learned), predicted from cortical positions via an MLP (position-based), or the same vector is shared across neurons, but can differ in scale (same). All models use a latent dimension $k = 12$. Standard error of the mean (SEM) is reported across three models with different initializations.

|  | Freely learned | Position based | Same |
| --- | --- | --- | --- |
| Log-Likelihood ↑ | **-0.34** ±0.05 | -0.69 ±0.04 | -0.88 ±0.04 |
| Conditioned Correlation ↑ | **0.24** ±0.003 | 0.22 ±0.005 | 0.18 ±0.003 |

strongly present in the data. As a control, we trained a model where all neurons shared the same feature vector, differing only by a per-neuron scale. Its worse performance shows that neurons differ not just in response strength, but also in which latent dimensions they emphasize across the cortex.

**Cortical-based latent mapping can extrapolate to unseen neurons**     To test if the cortical-based mapping from the previous experiment generalizes to unseen neurons, we removed $N = 500$ neurons per mouse during the first phase of training the latent model. In the second phase, we only finetune the weights and spatial position of $f_{it}^{(q)}$ and $f_{it}^{(\theta)}$ of the video-encoding part (Eq. (1)), and the latent feature vectors $\mathbf{w}_i^{(\theta)}$ and $\mathbf{w}_i^{(q)}$ are computed by the previously trained MLP. We compare the performance of the fine-tuned model on those $N = 500$ neurons against a model which was trained end-to-end on all neurons (Tab. 6 'Original'). We find that the position-based latent model can extrapolate the mapping of latents to unseen neurons using only their cortical position with almost no drop in predictive performance, demonstrating that the learned cortical-based mapping can extrapolate to new neurons using cortical coordinates alone.

Table 6: Correlation for 500 held-out neurons per mouse, conditioned on half of the training neurons ('Neurons'). We compare a model trained on all neurons ('Original') with a model that must extrapolate the mapping for the latents to those neurons ('Extrapolation'). SEM is across mice.

|          | Neurons | Original | Extrapolation |
|----------|---------|----------|---------------|
| **Mouse 6**  | 3932 | 0.185 | 0.185 |
| **Mouse 7**  | 3954 | 0.255 | 0.245 |
| **Mouse 8**  | 3970 | 0.205 | 0.205 |
| **Mouse 9**  | 4101 | 0.215 | 0.210 |
| **Mouse 10** | 4061 | 0.205 | 0.205 |
| **Average**  | 4003 | $0.213 \pm 0.005$ | $0.210 \pm 0.004$ |

## 5   Discussion

Cortical neuron activity variability arises primarily from two sources: stimulus-driven variability and internally driven variability from unobserved processes, such as behavioral tasks or brain states, which induce correlated activity across neurons. In this work, we showed that adding latent factors to a video encoding model enables the prediction of a joint dynamic response distribution and captures biological variables by implicitly learning correlations between latent factors and behavior. We also showed that the influence of the latent variables on the neuronal response is topographically organized in a robust way and that the latent variables are not primarily driven by the visual input.

**Limitations and possible extensions**     Although it remains difficult to isolate how much variance is explained by visual input alone [34], our control analyses suggest that the latent variables largely encode stimulus-independent factors; a residual stimulus-driven component, however, cannot be fully excluded. The model also uses more parameters per neuron than the video-only baseline. For simplicity, we implemented our latent method on a competition baseline [43]; future work should assess its performance when paired with state-of-the-art core representations.

Future extensions could relax distributional assumptions toward non-isotropic or non-Gaussian priors, replace the ZIG loss with a flow-based objective, or embed the method in new state-of-the-art transformer models [45]. Our method should integrate with most of the modern models that predict neuronal responses. It has to be further investigated whether the robust topographic organization of the latents reflects a biologically meaningful length scale, spatially organized brain waves [53], or simply residual retinotopic influences not captured by the video model.

**Acknowledgements**     We thank Dominik Becker, Michaela Vystrčilová, Lucas Mohrhagen, Florain Seifert, Alexander Ecker for the technical help and insightful discussions. Computing time was made available on the high-performance computers HLRN-IV at GWDG at the NHR Center NHR@Göttingen. The center is jointly supported by the Federal Ministry of Education and Research and the state governments participating in the NHR (www.nhr-verein.de/unsere-partner). FHS acknowledges the support of the German Research Foundation (DFG): SFB 1233, Robust Vision: Inference Principles and Neural Mechanisms – Project-ID 276693517 – and SFB 1456, Mathematics of Experiment – Project-ID 432680300, and the European Research Council (ERC) under the European Union's Horizon Europe research and innovation programme (Grant agreement No. 101171526). FHS acknowledges the support of the Lower Saxony Ministry of Science and Culture (MWK) with funds from the Volkswagen Foundation's zukunft.niedersachsen program (project name: CAIMed - Lower Saxony Center for Artificial Intelligence and Causal Methods in Medicine; grant number: ZN4257. This project has received funding from the European Research Council (ERC) under the European Union's Horizon Europe research and innovation programme (Grant agreement No. 101041669).

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

# A    Related Works Overview

An overview of related work, which either predicts neuron responses for given natural stimuli, uses latent representations for encoding neuron responses or takes behavior of the animal into account, is summarized in App. A.

Table 7: Summary of Reviewed Literature: **Neural activity** categorizes if the recorded neural activities are dynamic or static, and their role as input/output of the model. **Stimulus** details the type of stimulus components. **Learned Latent** indicates if the used model has a latent space and if a distribution is computable for the latent (probabilistic). **Behavior** describes the involvement and role of behavioral data. **Datasets** lists the datasets, including subject types, data collection methods, tasks performed during recordings, and neuron sample sizes.

| References | Neural activity | Stimulus | Learned latent | Behavior | Datasets |
|---|---|---|---|---|---|
| Schneider et al. [36] | Dynamic Input | No | Yes, probabilistic | Input | Rats, Mice, Monkey 2p and electro-physiology 10-1000 Neurons |
| Wang et al. [47] Turishcheva et al. [43] Sinz et al. [39] | Dynamic Output | Video | No | Input | Mice, 2p, passive $\sim$ 140,000 Neurons $\sim$ 40,000 Neurons |
| Kim et al. [25] | Dynamic In/Output | Audio | Yes, probabilistic | No | Synthetic and Rats, audio decision-making 67 Neurons |
| Antoniades et al. [3] Azabou et al. [4] | Dynamic Output | Video No | Yes, not probabilistic | Output | Mice, 2p, passive 386 Neurons Monkey 27,373 Neurons |
| Gokcen et al. [20] Sussillo et al. [41] | Dynamic Input | No | Yes, probabilistic | No | Macaque V1-3 120 Neurons Synthetic 30 Neurons |
| Zhou and Wei [56] Wang et al. [48] Bjerke et al. [6] Jensen et al. [24] | Dynamic In/Output | No | Yes, probabilistic | Task Input No No | Monkey, reaching-task Rat, running 192,120 Neurons Monkey, Rat task 200 x 200 Neurons Mouse,Rats 26 and 149 Neurons Macaque, reaching-task 200 Neurons |
| Geenjaar et al. [18] | Dynamic In/Output | No | Yes not, probabilistic | No | fMRI |
| Seeliger et al. [37] | Dynamic Output | Video | No | No | fMRI |
| Bashiri et al. [5] | Static Output | Image | Yes, probabilistic | Output | Mouse V1/LM, 2p,passive $\sim$ 4,000 Neurons |

## B  Monte-Carlo Approximations

We have to approximate expectation values via Monte-Carlo sampling two times in our model setup. First, during training we have to calculate the likelihood $\langle \log p(\mathbf{y}|\mathbf{z}, \mathbf{x}) \rangle_{\mathbf{z} \sim q_\phi(\mathbf{z}|\mathbf{y})}$ in the ELBO via Monte-Carlo sampling (Eq. (3)). Second, during evaluation, we draw samples from the prior distribution $\mathcal{N}(0, \mathbf{I}_k)$, where $k$ is the latent dimension, for calculating the log-likelihood $\log p(\mathbf{y}|\mathbf{x})$ as described in Eq. (2).

The ELBO was approximated with up to 150 samples. More samples yield memory issues during training on a single A100 GPU. We observed the log-likelihood performance during inference to increase with the sample size during training as the ELBO is approximated more accurately (Fig. 6). For the Monte-Carlo approximation of the marginalized log-likelihood during inference, we tested different sample sizes ranging up to 5000. It was not possible to test more than 5000 samples, because at that point we ran into memory issue on a single A100 GPU. However, Fig. 7 indicates that the log-likelihood stabilizes and approaches a plateau after approximately 1000 samples. We thus resorted to drawing 1000 samples to approximate the marginalized log-likelihood. The correlation performance of the latent model was not strongly influenced by the sample sizes.

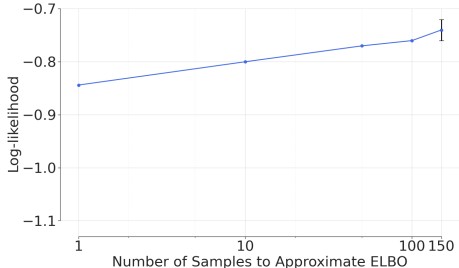 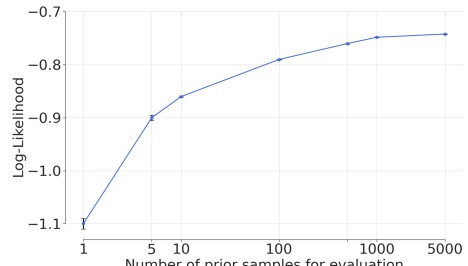

Figure 6: Log-likelihood (measured in bits per neuron and time) of models trained with different sample sizes to approximate the ELBO during training. The error bar indicates the standard error of the mean of the log-likelihood for models initialized and trained on 3 different seeds. Due to the computational cost of training a model, the standard error was calculated only at a single point.

Figure 7: Log-likelihood (measured in bits per neuron and time) for different sample sizes $L$ of the prior during evaluation. Error bars indicate the standard error of the mean for sampling from different seeds.

## C  Model Architecture and Hyperparmeters Setting

All experiments and training of models were done on a single A100 GPU with 80GB of memory.

The video processing part consists of a factorized 3D-CNN block followed by a Gaussian readout ([23]). Each convolutional layer consists of a factorized 3D convolution across spatial and temporal dimensions followed by a batch normalization layer and an ELU activation function. We use a variational autoencoding approach for the latent representations. A dropout layer is applied to the neuron responses before they are fed into the encoder. This prevents the model from learning correlations between specific neurons, thereby encouraging the learning of global latent representations. The encoder applies a linear layer reducing the dimensionality of neuron responses $N \approx 8,000$ ($N$ ranging from 7800 to 8200

Table 8: Hyperparameter Configuration

| General | |
| --- | --- |
| Learning rate | 0.005 |
| **Core** | |
| Number Layers | 3 |
| Temporal Kernel Size first Layer | 11 |
| Spatial Kernel Size first Layer | (11,11) |
| Spatial Kernel Size other Layer | (5,5) |
| Spatial Kernel Size other Layer | (5,5) |
| Channels per Layer | (32,64,128) |
| **Encoder** | GRU |
| Number Layers | 1 |
| Dropout probability | 50% |
| Output Dim Linear Layer | 42 |
| Output Dim GRU | 12 |

Table 10: An overview of the details of each experiment conducted in the main text. In the row for Table 1 we only refer to the latent model.

|  | Positional Data | Pre-trained Video-Part | Latent Dimension $k$ | Evaluation Neurons (per mouse) | Used Mice ID'S in 12 |
|---|---|---|---|---|---|
| Table 1 | True | True | 12 | Half | 6-10 |
| Table 2 | False | True | 12 | Half | 1-5 |
| Table 3 | True | True | 12 | Half | 1-5 |
| Table 4 | True | True | 12 | Half | 6-10 |
| Table 5 | False | False | 12 | 500 | 6-10 |
| Figure 3 | True | True | 1-200 | Quarter | 6-10 |
| Figure 4 | True | True | 1-200 | Quarter | 6-10 |
| Figure 5 | True | True | 12 | All | 9 |

depending on the mouse as part of the SENSORIUM dataset [44]) to $n \ll N$. For each time step, the same linear layer is applied independently. The linear layer is followed by a layer normalization and an ELU activation function. An individual linear layer was trained for each mouse. The output is processed by a one-layer recurrent neural network with gated recurrent units producing the means $\mu(\mathbf{y}; \phi)$ of the approximate posterior $q(\mathbf{z}|\mathbf{y}; \phi) = \mathcal{N}(\mu(\mathbf{y}; \psi), \sigma^2\mathbf{I})$. $\sigma$ is a learnable model parameter of the encoder. $\mathbf{z}$ is combined with the outputs from the video encoding part computing the parameters for the probability $\log p(\mathbf{y}|\mathbf{z}, \mathbf{x})$ as in Eq. (1). For a graphical illustration, see Fig. 2. We searched over various hyperparameters (Table 8) using Optuna [1].

## D  Optimizing Encoder Architecture

To ensure optimal encoding of the neuron responses, we tested a 3D factorized CNN and a GRU architecture. We further tested different layer sizes ranging from 1-5. We found that a one-layer GRU encoder performs best in terms of correlation and log-likelihood.

Further, we tried to add a "smoother" $g$ to further smooth the latents in time. Compared to the model reported above, $\mathbf{z}_t$ is replaced by $g(\mathbf{z}_t)$ in Eq. (1). We tested how a model with a CNN-smoother or GRU-smoother model performed compared to a model without a smoother. As Tab. 9 indicates, replacing the GRU with a CNN does not make a significant difference. However, the model without smoother had the best performance in log-likelihood and conditioned correlation (Tab. 9).

## E  Experiment Details

Depending on the analysis setup, we adjusted the model architecture and training configuration as needed. For experiments examining cortical patterns revealed by the latent model, we ensured that no positional data was provided. Although the Gaussian readout allows neuron positions to be used for spatial predictions in the video feature map, we disabled this option for these experiments (Tab. 10, 'Positional Data' column). Additionally, we found that initializing the video-processing weights of our latent model with those from a pre-trained video-only ZIG model improved predictive performance (Tab. 11). Therefore, we applied pretrained weights for the video component whenever the experiment setup allowed (Tab. 10, 'Pretrained Video-Part' column). Since we tested different latent dimensions, we specify the latent dimensions used for each experiment (Tab. 10, 'Latent Dimension' column). Lastly, we report the number of neurons used during evaluation (Tab. 10, 'Evaluation Neurons' column). If the same number of neurons was used for evaluation across experiments, the selected neurons remained consistent. For instance, if one-quarter of the neurons were evaluated, the same subset was used in all experiments. For an exact number of neurons per mouse, see Tab. 12.

Table 9: The columns present the performance of models with a 12-dimensional latent space, each trained using either a GRU smoother, a CNN smoother, or no smoother.

|  | GRU Smoother | CNN Smoother | No Smoother |
|---|---|---|---|
| Log-Likelihood ↑ | -0.74 | -0.77 | **-0.30** |
| Conditioned Correlation ↑ | 0.23 | 0.22 | **0.24** |

Table 11: Comparison of a latent model with latent dimension $k = 12$, initialized using weights from a pre-trained video-only ZIG model, against a latent model trained from scratch. We evaluate performance based on log-likelihood and correlation, considering both Conditioned Correlation (conditioning on half of the neurons) and Prior Correlation (sampling from the prior). Evaluation is conducted on half of the neuron population.

| | Pre-trained | Scratch |
|---|---|---|
| Log-Likelihood ↑ | **-0.30** | -0.36 |
| Conditioned Correlation ↑ | **0.24** | 0.23 |
| Non-Conditioned Correlation ↑ | **0.18** | 0.10 |

# F   Mice IDs

For reproducibility, we list the Sensorium ID's of the mice, which were used for training and evaluation in our experiments, in Tab. 12. In the main part, they were referred to as mice 1-10. Further, Tab. 12 shows the exact number of recorded neurons per mouse.

Table 12: Mice 1-10 of the main text with corresponding SENSORIUM IDs ('IDs'). The total numbers of recorded neurons per mouse are tracked ('Neurons').

| | Neurons | ID |
|---|---|---|
| **Mouse 1** | 7440 | 29156-11-10 |
| **Mouse 2** | 7928 | 29228-2-10 |
| **Mouse 3** | 8285 | 29234-6-9 |
| **Mouse 4** | 7671 | 29513-3-5 |
| **Mouse 5** | 7495 | 29514-2-9 |
| **Mouse 6** | 7863 | 29515-10-12 |
| **Mouse 7** | 7908 | 29623-4-9 |
| **Mouse 8** | 7939 | 29712-5-9 |
| **Mouse 9** | 8202 | 29647-19-8 |
| **Mouse 10** | 8122 | 29755-2-8 |

# G   Trade-off between Modeling distribution and Modeling conditional Mean

As the ZIG distribution is not an exponential family with the data mean as a sufficient statistic, better modeling of the full response distribution (i.e., higher log-likelihood) does not necessarily translate into more accurate conditional-mean predictions (i.e., higher correlation).

To check this, we trained a ZIG model using a Poisson loss during training. For this, we insert the means of the predicted ZIG distributions into the Poisson loss. For each neuron $i$ and timepoint $t$, our ZIG model predicts the ZIG-parameters $q_{it}(\mathbf{x}; \psi)$ and $\theta_{it}(\mathbf{x}; \psi)$ (as in Fig. 2 A) based on the video input $\mathbf{x}$ and its parameters $\psi$. With these we analytically compute the mean of the ZIG distribution $\hat{r}_{it}(q_{it}(\mathbf{x}; \psi), \theta_{it}(\mathbf{x}; \psi), \kappa_i, \rho)$. During training, we then use Poisson Loss between those predicted responses $\hat{r}_{it}$ and the observed neural responses $r_{it}$ as objective:

$$\min_{\psi} \sum_{it} \hat{r}_{it}\left( q_{it}(\mathbf{x}; \psi), \theta_{it}(\mathbf{x}; \psi), \kappa_i, \rho \right) - r_{it} \cdot \log\left( \hat{r}_{it}(q_{it}(\mathbf{x}; \psi), \theta_{it}(\mathbf{x}; \psi), \kappa_i, \rho) \right)$$

Consistent with our hypothesis above, as shown in Tab. 13, training our video-only ZIG model with a Poisson loss raises its correlation from 0.183 to 0.195, comparable to the Poisson baseline model.

We observe a similar pattern with our latent model: swapping its training objective from ZIG to Poisson loss yields a notable boost in correlation (especially when conditioning on half of the neuronal population) (Tab. 13).

Table 13: Predictive performance of the Poisson Baseline model, a video-only ZIG model trained with Poisson loss, and a video-only ZIG model trained with ZIG loss. Further, the predictive performance of a Latent ZIG model trained on Poisson-loss and ZIG-loss is reported with and without conditioning on half of the neurons' responses. Log-likelihood is measured in bits per neuron and time. Performance is evaluated only on half of the neurons that are not used for conditioning. Within each scenario—"Non-Conditioned" and "Conditioned"—the best result is set in boldface.

| | Poisson Baseline | ZIG Poisson-loss | ZIG ZIG-loss | Non-Conditioned | | Conditioned | |
| --- | --- | --- | --- | --- | --- | --- | --- |
| | | | | Latent Poisson-loss | Latent ZIG-loss | Latent Poisson-loss | Latent ZIG-loss |
| Pearson Correlation ↑ | **0.195** | **0.195** | 0.183 | 0.184 | 0.183 | **0.266** | 0.230 |

# H  Flow Model

[5] successfully introduced a flow to enhance predictive capabilities of their model for neural data [14]. Hence, we tested, how adding a flow $T$ applied on our neural responses $\mathbf{y}$ impacts our model.

We used a Gaussian distribution as base distribution which is independent across time and neurons. A Gaussian distribution has the advantage, that we do not have to restrict the transformed responses $T(\mathbf{y})$ to be positive. To capture the peak at zero, the responses below the threshold $\rho$ are modeled by a uniform distribution, while the flow is used to capture responses above the threshold. We model the base distribution independently over time. Thus, for a given latent space $\mathbf{z}_t$ and video $\mathbf{x}$, the likelihood of the transformed neuron responses $T(\mathbf{y}_t)$ at time point $t$ is given by:

$$p(T(\mathbf{y}_t)|\mathbf{z}_t, \mathbf{x}) = \prod_{\{i:y_{it}<\rho\}} \left( \frac{1 - q_{it}(\mathbf{x}, \mathbf{z}_t)}{\rho} \right) \cdot \prod_{\{i:y_{it}\geq\rho\}} \left( q_{it}(\mathbf{x}, \mathbf{z}_t) \cdot \mathcal{N}\big(\mu_i(\mathbf{x}, \mathbf{z}_t), I\big) \right)$$

The flow model consists of two parts. (1) A flow model $T$, with learnable parameters, that transforms the neural responses $\mathbf{y}_t$ for each time point in the same way. The flow acts independently across the $N$ neurons: $T(\mathbf{y}_t) = [T_1(y_{1t}), T_2(y_{2t}), \ldots, T_N(y_{Nt})]$ (2) A latent space model, as described in Sec. 3, which predicts the response distribution of the transformed responses $\mathbf{r}_t := T(\mathbf{y}_t)$. Thus, the expected value $\mu_i(\mathbf{x}, \mathbf{z}_t)$ and response likelihood $q_{it}(\mathbf{z}_t, \mathbf{x})$ are predicted by our latent space model. To determine the actual response likelihood $p(\mathbf{y}_t|\mathbf{z}_t, \mathbf{x})$, we use the change of variable formula:

$$p(\mathbf{y}|\mathbf{z}, \mathbf{x}) = p(T(\mathbf{y})|\mathbf{z}, \mathbf{x}) \cdot \big| \det \nabla_y T(\mathbf{y}) \big|$$

As we choose to apply the flow on each neuron and time dimension separately, this results in a diagonal Jacobian:

$$\det \nabla_y T(\mathbf{y}) = \prod_{it} \frac{\partial T_i(y_{it})}{\partial y_{it}}$$

To predict a response for neuron $i$ at time point $t$, we compute the mean of the random variable $y_{it} = T_i^{-1}(r_{it})$ via Monte-Carlo sampling:

$$\mathbb{E}[T_i^{-1}(r_{it})] = \int T_i^{-1}(r_{it})\, p(r_{it}|\mathbf{x}, \mathbf{z}_t)\, dr_{it} \approx \frac{1}{M} \sum_m T_i^{-1}(r_{it}^{(m)})$$

For the experiments, we used a simple flow consisting of two functions:

$$T = \text{affine} \circ \text{softplus}^{-1}$$

The affine layer has a learnable scale parameter $a_i$ and bias parameter $b_i$ for each neuron. It computes $\text{affine}(y_i) = ay_i + b_i$. The inverse softplus layer is used to make sure that response values $T^{-1}(\hat{y})$ are positive for a sample $\hat{y}$ from the base distribution.

We tested different flows consisting of more functions, where a affine function and a non-linear

function alternated. We tested log, tanh, exp, ELU as non-linear functions. For example, we experimented with the flow used in [5]:

$$T = \text{affine} \circ \exp \circ \text{affine} \circ \text{ELU} \circ \text{affine} \circ \text{ELU} \circ \text{affine} \circ \log \circ \text{affine}$$

However, all those models either collapsed or could not improve the performance of the first tested simple flow.

Tab. 14 shows, that a model with a simple flow indeed improved the average log-likelihood compared to a base model, which was trained without a flow ($T = \text{identity}$) using a Gaussian base distribution $p(y|x,z) = \mathcal{N}(y; \mu(z,x), I)$ for non-zero responses. However, the default latent space, which models ZIG distributions, has still the highest log-likelihood of -0.74 bits per time and neuron. When the encoder sees half of the neurons, the ZIG model slightly outperformed the Gaussian model with and without a flow in terms of correlation (Tab. 14).

Table 14: Comparison between a default latent ZIG model, a Gaussian latent model and a gaussian latent model with a flow

| Model | ZIG Model | Gaussian Model | Flow Gaussian |
|---|---|---|---|
| Log-likelihood in Bits per Neuron and Time | **-0.88** | -2.1 | -1.2 |
| Conditioning on half of neurons | **0.229** | 0.219 | 0.211 |

## I  Comparison with a Vision Transformer Core

To test the transferability of our latent method, we repeated the main comparison using a Vision Transformer (ViT) core following Li et al. [28]. The trends are qualitatively the same as with the 3D CNN core: the ZIG model with the ViT core performs slightly worse than the Poisson baseline, consistent with the optimization trade-off between modeling full distributions and conditional means discussed in the main text. In contrast, the latent model improves the performance beyond the Poisson baseline, demonstrating that our approach generalizes across different core architectures.

Table 15: Performance comparison using a Vision Transformer (ViT) core.

| | Poisson Baseline | Video-only ZIG | Latent ZIG |
|---|---|---|---|
| Correlation ↑ | **0.17** (0.007) | 0.155 (0.009) | 0.153 (0.007) |
| Log-Likelihood ↑ | – | -1.2 (0.09) | **-0.45** (0.06) |

While we did not obtain state-of-the-art performance with ViT—likely due to limited hyperparameter tuning—the results indicate that the latent method we propose is readily transferable to other cores.

Additionally, conditioning on half of the neurons further boosts the correlation up to 0.194 (0.006), showing that encoding neuron responses also works effectively with the ViT core.

## J  Models with Behavior

Table 16: Predictive performance of a Poisson Baseline model, a video-only ZIG-model with behavior and a latent model without behavior.

| | Poisson Baseline | Video-only ZIG | Latent |
|---|---|---|---|
| Conditioned Correlation ↑ | 0.20 | 0.19 | **0.27** |
| Log-Likelihood ↑ | – | –0.94 | **0.18** |

To assess whether measured behavior could substitute for our inferred latent state, we augmented both the Poisson baseline and the video-only ZIG model with the two measured behavioral variables: treadmill speed and pupil dilation. These behavioral variables were concatenated as additional input channels into each model's core as in [43, 47]. As reported in Tab. 16, the latent state model (with latent dimension $k = 150$, and conditioned on half of the neurons held out from evaluation) surpasses both behavior-augmented models in terms of correlation and log-likelihood. Behavior was not included for the latent model.

## K   Heat Maps of Response Distributions

As reported in the main paper, a high-dimensional latent state yields a higher log-likelihood. We explained this effect by a better "coverage" of the response space with probability mass. Here, we visualize this effect. One can see that a high-dimensional latent model is more likely to put some probability mass on a broader range of response values and does not concentrate all the probability mass around the mean (Fig. 8). Thus, extreme response values, as in column 3 of Fig. 8, are covered with a non-zero probability. A low-dimensional model predicts them with almost zero probability (parts of the response traces are in the gray areas of the heatmap).

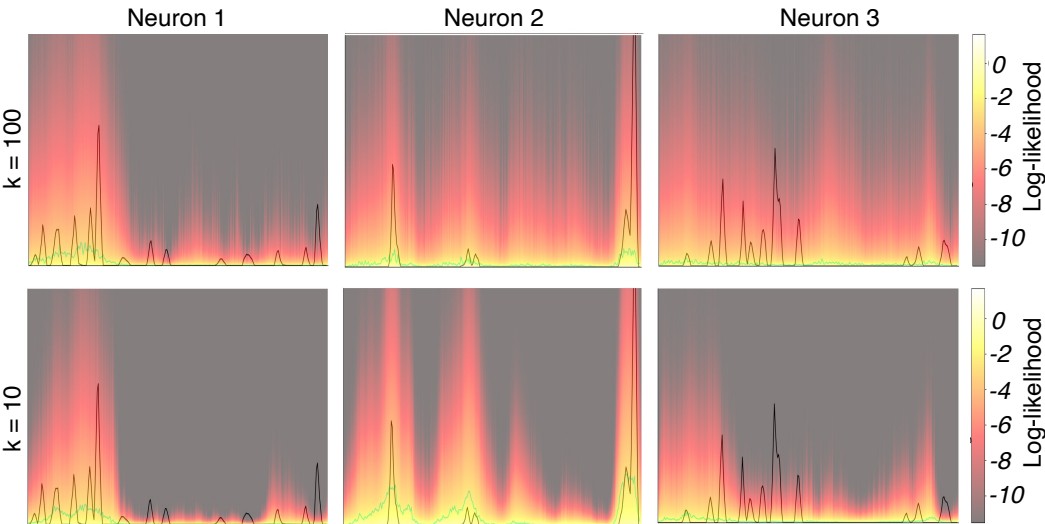

Figure 8: Neuron response traces and model predictions for one video recording. The black line represents the actual response of each neuron over time, while the cyan line represents the model's predicted response, which is the mean of the predicted ZIG distribution. For each time point, the predicted response ZIG-distribution is visualized as a heat map along the y-axis. The top row displays predictions from a model with a latent dimension of $k = 100$, while the bottom row shows predictions from a model with a latent dimension of $k = 10$.

## L   Behavior Analysis

For the pupil dilation and the treadmill speed of each mouse, we performed one CCA analysis each. The CCA analysis was done with 5-fold cross-validation. We split the recording time with an 80/20 ratio and fit the CCA weights on the train split. The correlations were computed between the CCA combination $\sum_i w_i^{(cca)} \mathbf{z}^{(i)}$ and the behavioural variable (pupil dilation or treadmill speed) on the test split. $w_i^{(cca)}$ are the CCA weights and $\mathbf{z}^{(1)}, \ldots, \mathbf{z}^{(k)} \in \mathbb{R}^T$ are the latent variables. For each mouse, we computed the average correlation across the cross-validation. For two selected videos and mice, we plotted their normalized pupil dilation and treadmill speed against the corresponding normalized CCA combination of the latent over the whole video time, which corresponds to $\sim 300$ time points (Fig. 9).

**CEBRA baseline explained.** To compare how good our models is we used CEBRA [36] as a baseline. CEBRA performs dimensionality reduction on neural activity using InfoNCE contrastive learning, where positive and negative pairs are defined by auxiliary variables such as time or behavior.

Table 17: CEBRA hyperparameter search (Optuna) and selected setting used for all mice.

| Hyperparameter | Search range | Selected value |
|---|---|---|
| Time offset | 5–20 (integer) | **6** |
| Batch size | $\{32, 64, 128, 256, 512, 1024\}$ | **64** |
| Learning rate | $[10^{-4}, 10^{-2}]$ (log-uniform) | **0.003** |
| Latent dimension | 2–32 (integer) | **24** |

When the auxiliary variable is discrete, for example a left or right wheel turn, it selects positives uniformly from all samples with the same label. When the variable is continuous, such as running speed or pupil direction, it chooses a random point within a time window around the sample and then find the closest match in the dataset using either Euclidean or cosine distance; this sample becomes the positive pair, which adds diversity and prevents repeatedly selecting the same example. Negative pairs are sampled randomly. For decoding, CEBRA encode neural responses, find the nearest latent vectors for responses in the training set, and returns their associated behavioral variables as predictions.

**CEBRA hyperparameters tuning.** For each mouse, we fitted one CEBRA model independently and computed latent–behavior correlations using the same CCA protocol and the same train/validation split as for our latents. We tuned CEBRA's hyperparameters with Bayesian optimization (Optuna), exploring batch size, latent dimensionality, learning rate, and time offset, and selected the setting that maximized the average validation behavior correlation. The search ranges and the selected values (used for all mice) are summarized in Tab. 17.

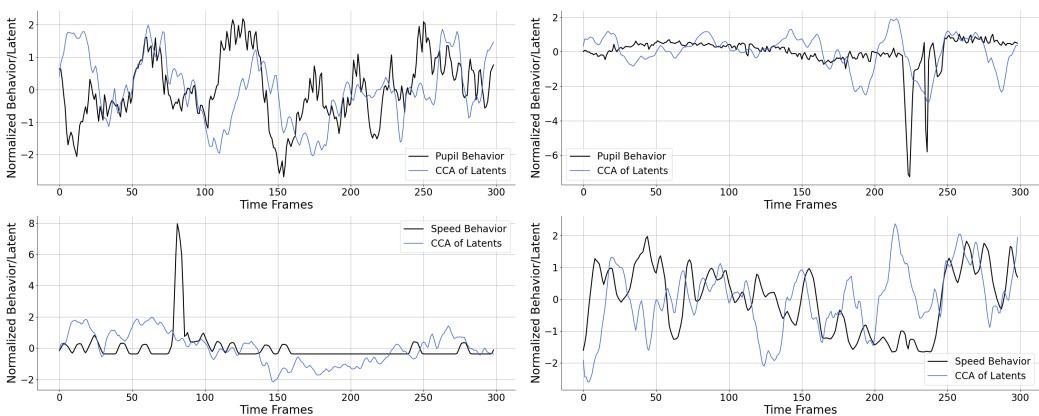

Figure 9: Normalized behavior and CCA combination of latent factors during two selected videos

# M   Additional Brain Maps

In Fig. 5, we showed only the first three singular dimensions of the latent feature matrix $\mathbf{w}^{(q)}$. In rows 1 and 2 of Fig. 10, we compare those same three dimensions for both $\mathbf{w}^{(\theta)}$ and $\mathbf{w}^{(q)}$. They exhibit largely the same spatial patterns, aside from sign flips in columns 1 and 3. Beyond the first three dimensions, we did not observe any notable structure: Row 2-4 of Fig. 10 displays singular dimensions one through nine for $\mathbf{w}^{(\theta)}$. Beyond the fourth singular dimension, they do not appear to align with cortical neuron positions and seem randomly distributed across the xy-plane.

# N   Latent model performance on out-of-distribution data

For the training of the latent model, we usually initialize the weights of the core, which encodes the video input, with the weights of a pretrained video-only ZIG model, which doesn't have a latent state. If we train the latent model from scratch, the model seems to over-rely on the information of the latent state, which encodes responses of half of the neurons' population during training. When this information is marginalized out ('Non-Conditioned'), the model performs poorly in terms of correlation Tab. 18. However, this can be prevented by loading a pretrained core ensuring that the latent state mostly encodes additional information, which is not contained in the output of the core Tab. 19. When conditioning on half of the neurons' responses, both training procedures yield comparable results (Tab. 19,Tab. 18 'Conditioned').

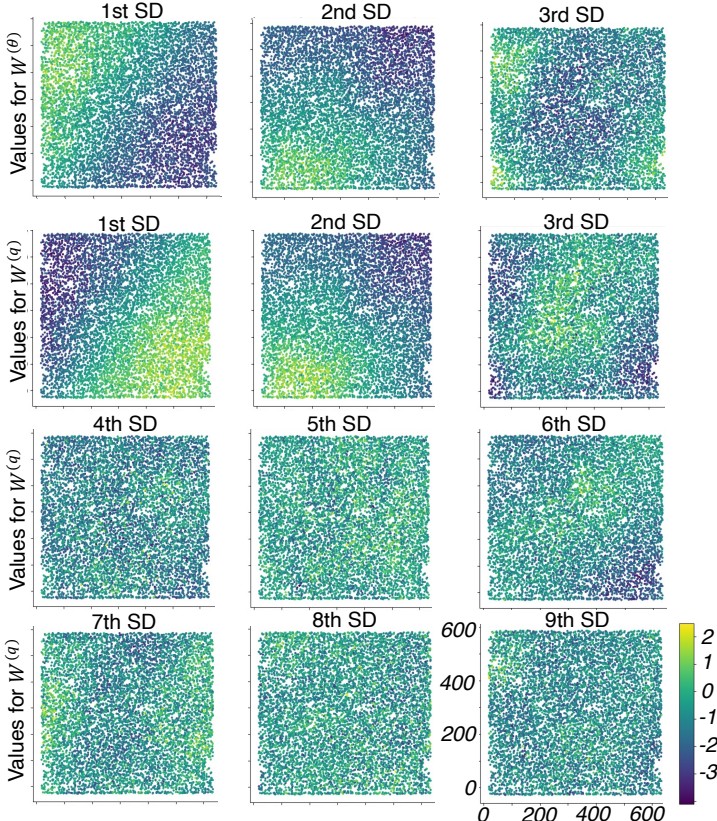

Figure 10: Row 2-4: Color gradient maps of the latent feature vectors $\mathbf{w}_i^{(q)}$ along the first nine Singular Dimensions (SD) for mouse 4. Row 1: The color gradient maps of the first three singular dimensions of the latent feature vectors $\mathbf{w}_i^{(\theta)}$ are plotted. All recorded neurons are located within a $600\mu\text{m} \times 600\mu\text{m}$ square in the cortex. Their depth differs at most $200\ \mu\text{m}$. The model was trained without any knowledge of cortical positions and freely learnable feature vectors.

Table 18: Latent model without ZIG pretraining, $k = 200$

|          | Non-Conditioned | Conditioned |
|----------|-----------------|-------------|
| **Mouse 6**  | 0.016 | 0.170 |
| **Mouse 7**  | 0.029 | 0.196 |
| **Mouse 8**  | 0.022 | 0.211 |
| **Mouse 9**  | 0.026 | 0.209 |
| **Mouse 10** | 0.015 | 0.184 |

Table 19: Latent model with ZIG pretraining, $k = 150$

|          | Non-Conditioned | Conditioned |
|----------|-----------------|-------------|
| **Mouse 6**  | 0.086 | 0.172 |
| **Mouse 7**  | 0.121 | 0.192 |
| **Mouse 8**  | 0.125 | 0.206 |
| **Mouse 9**  | 0.107 | 0.207 |
| **Mouse 10** | 0.082 | 0.179 |

## O   Broader impact

Accurate models of neural variability deepen our insight into how brains transform sensory information and may shed light on the disruptions underlying neurological disorders. Specifically, a more precise model that integrates internal brain states, stimulus-driven activity, and anatomical structures such as retinotopy or memberships to certain brain areas could uncover deeper insights into the cortex's computational principles. Although training our models still depends on animal data, we rely on existing, broadly applicable datasets to maximize the scientific yield of each experiment. Moreover, approaches like the model presented here can help to reduce the number of animal experiments by enabling researchers to explore functional principles of brain processes with faithful models *in silico*.

