# OpenReview forum: "Modeling Dynamic Neural Activity by combining Naturalistic Video Stimuli and Stimulus-independent Latent Factors"
_NeurIPS.cc/2025/Conference — NeurIPS 2025 poster_

### Official Review · Reviewer_cFTH · 2025-06-26

**Clarity:** 2
**Significance:** 3
**Originality:** 3
**Rating:** 4
**Confidence:** 2

**Summary:**

This paper proposes a novel algorithm for predicting neural responses to dynamic naturalistic stimuli while modeling stimulus-independent variability in the neuronal population (internal brain states).

Most previous approaches tend to either predict neural activity from stimuli without modeling a latent state or focus on modeling a latent state from neural responses. Some previous work predicts neural activity based on sensory and neuronal input. However, the only previous work combining visual input with a latent state is restricted to static stimuli.

To bridge this gap, the authors propose a model to predict neural activity combining video visual input with a latent state. Technically, it consists of a 3D convolutional neural network “core” that processes the video frames and a latent variable that could be conditioned upon neural activity and captures stimulus-independent fluctuations. The response of each neuron is modeled with a Zero-Inflated Gamma (ZIG) distribution whose parameters depend on both the video-driven features and the latent state. ZIG is chosen because it aligns better with calcium imaging data.

The model is evaluated on neural recordings from mouse primary visual cortex (V1) in response to grayscale natural video stimuli (from the SENSORIUM dataset). The proposed video+latent model improves the log-likelihood of the neural responses over baselines and also achieves comparable Pearson correlation. In addition, the proposed method seems to handle out-of-distribution stimuli better.

More importantly, the latent states highly correlate with behavioral variables (pupil dilation and locomotion speed), although they are not used in training. This suggests the latent state successfully captured intrinsic neural dynamics. The latent states also seem to exhibit topographic organization, although the cortical positions of the neurons are not used in the training. Those conclusions show that the latent features uncovered by the model have a meaningful biological basis.

**Questions:**

1. Regarding weakness #2: Would you be able to evaluate the quality of the latent factors against a baseline? I am impressed that they have a solid biological basis, but I am wondering if they are somehow better than a baseline because they are learned together in a neural activity prediction task.
2. Mouse v1 neurons should encode other features such as orientation. Your analysis in table 3 shows that the latents encode contrast minimally. Do you think it'd be meaningful to extend your analysis to other features?
3. For the neurons that are conditioned upon - how are they chosen? I would assume they are chosen randomly from reading the paper. Does using a different subset of neurons matter? Does restricting the choice to purely layer 2/3/4/5 matter? I am wondering if the different choice here can lead to "better" latent states or higher accuracy.

**Ethical Concerns:**

["NO or VERY MINOR ethics concerns only"]

**Final Justification:**

I appreciate that the authors have conducted additional experiments to address my concerns and curiosity. Overall, this is a technically solid paper, and the results are interesting; I believe this is a meaningful contribution to the research community.

**Limitations:**

yes

**Quality:**

3

**Strengths And Weaknesses:**

Strengths:
1. The paper proposes a novel integration of dynamic visual stimuli and latent brain states in order to predict neural activity, addressing a gap in the literature.
2. The latent states (learned in an unsupervised fashion during training) have a meaningful biological basis. They highly correlate with behavioral variables, and they reflect an anatomical organization.
3. The model handles OOD samples better
3. The paper conducted a thorough experiment and analysis

Weaknesses:
1. The proposed model (more complex than the baseline) does not substantially improve (and in fact slightly underperforms) the baseline in terms of predicting the mean neural response as measured by correlation. The authors have provided an explanation for this, but I am not sure if the method's added complexity and parameters.
2. To remedy the above weakness, maybe the authors can show that the latent states (the other important part learned in the model) are somehow "better" than a baseline. But such a baseline is missing from the paper.

---

> ### Author Rebuttal · Authors · 2025-07-31
>
> Thank you for your positive review, highlighting the novelty of our work and our “thorough experiments and analysis”.
>
> Below we address your questions, and we will incorporate these replies into our manuscript:
>
>
> **Q1**:  You raise a valid point about baselines for the latent factors. The most relevant and commonly used "baseline" for internal brain states are observable behavioral variables, and we performed this exact comparison. However, to address your question, we compared it with CEBRA  (Schneider et al., 2023). Please note, that CEBRA does not take visual stimuli as input or have held-out neurons. We trained CEBRA, which has the same latent dimension ($k=12$) as our model, on our data by fitting for each mouse one CEBRA model. Then, we performed the same canonical correlation analysis (CCA) with the CEBRA latents as we did for our latents (lines 322-327). “  Specifically, CCA “finds the linear combination of the latent variables $\mathbf{z_{(1,\dots,T)}}^{(1)},\dots,\mathbf{z_{(1,\dots,T)}}^{(k)}$ with maximal correlation to a chosen behavioral variable like pupil dilation or treadmill speed.”  The CCA analysis was done with 5-fold cross-validation in an 80/20 split.
> Below, we can see the canonical correlation with pupil dilation and treadmill speed on the test set. The error bars indicate the standard error of the mean of cross-validation. Especially for treadmill speed, the latents of our model do correlate stronger with behavior compared to the CEBRA latents, probably because explicitly separating stimulus information actually helps.
>
> |  Canonical Correlation with pupil dilation  | CEBRA    | ZIG latent
> |-|-|-
> |Mouse 1  |   0.65 (0.003)  | 0.66 (0.007) |
> |Mouse 2  |   0.48 (0.008)  | **0.59** (0.009) |
> |Mouse 3  |   0.52 (0.007)  |**0.64** (0.03) |
> |Mouse 4  |   0.59 (0.004)  | 0.59 (0.003) |
> |Mouse 5  |   **0.85** (0.001)  | 0.60 (0.01) |
> |**Average**  |   0.62 (0.005)  | 0.62 (0.01) |
>
>
> |  Canonical Correlation with treadmill speed  | CEBRA    | ZIG latent
> |-|-|-
> |Mouse 1  |   0.19 (0.003)  |**0.26** (0.02) |
> |Mouse 2  |   0.48 (0.008)  |**0.61** (0.02) |
> |Mouse 3  |   0.35 (0.003)  | **0.57** (0.005) |
> |Mouse 4  |   0.26 (0.01)  | **0.33** (0.03) |
> |Mouse 5  |   0.26 (0.007)  | **0.44** (0.007) |
> |**Average**  |   0.24 (0.005)  | **0.49** (0.02) |
>
> **Q2**: Yes, you are right that mouse V1 has orientation-selective neurons (as well as phase-selective neurons). However, this information should be encoded in the model's core and readout weights and not in the latents [1] as it is a function of the neuron that does not change from trial to trial, changing the response based on the stimuli.
> However, the key idea of analysis in Table 3 is to show that latents are mainly encoding stimuli independent information. This is expected from the model and we wanted to test against the case that the model uses the latent state to explain variability that it cannot explain with the video-driven part. In theory, it could also do that for orientation, but we’d expect this to be more complex. In addition, orientation is not as easily extracted from natural video as local contrast. We thus focused on contrast because it's a fundamental, low-level stimulus property that can be directly computed from the video frames. This made it a straightforward control that latents do not encode contrast (e.g., stimulus) information - the low R² values in Table 3 confirm this.
> To further probe stimulus independence beyond contrast, we ran an additional test: we predicted the held‑out neurons for trial A while conditioning on the responses of trial B (same movie, different brain state). If latents mainly carry stimulus information, performance should be unaffected, as stimulus does not change; but if they carry brain-state‑specific information, accuracy should fall. We observed the decrease in performance (Fig. 3, lines 300‑315), confirming that the latents represent a video‑independent brain state.
>
> **Q3**: Yes, your assumption is correct, and they are chosen randomly. We’ll clarify this in our final version. To answer your question, we ran additional experiments where we masked neurons based on their cortical coordinates. To test how sensitive the model is to that choice, we ran three experiments: we sorted the neurons by their cortical x, y, or z coordinate and then masked the first 50 % of each ordered list. Masking by x or y (cortical surface) caused a small drop in performance, whereas masking by z (cortical depth) left accuracy unchanged relative to random masking. This mirrors Fig. 5: latent patterns show clear topographic organisation across the cortical surface but remain largely constant with depth.
> Below are the results of the different masking strategies, we reported the standard error of the mean across the five mice.
> |    | Random    | x-coordinate      | y-coordinate     | z-coordinate      |
> |-|-|-|-|-
> |Conditioned Correlation  |   0.24 (0.002)  | 0.225 (0.003) | 0.22 (0.003)    |  0.24 (0.002)   |
>
> We will incorporate this analysis into the appendix.
>
> We hope that these additional comments and experiments will help you to understand our method better and revisit the evaluation.
>
> References:
>
> [1] Ustyuzhaninov, I., Burg, M. F., Cadena, S. A., Fu, J., Muhammad, T., Ponder, K., Froudarakis, E., Ding, Z., Bethge, M., Tolias, A. S., & Ecker, A. S. (2022, February 10). Digital twin reveals combinatorial code of non‑linear computations in the mouse primary visual cortex [Preprint]. bioRxiv. https://doi.org/10.1101/2022.02.10.479884

---

> > ### Comment · Reviewer_cFTH · 2025-08-01
> >
> > Thank you for addressing my questions and conducting additional experiments. I do not have follow-up questions for now.

---

### Official Review · Reviewer_WMJZ · 2025-06-30

**Clarity:** 3
**Significance:** 3
**Originality:** 2
**Rating:** 4
**Confidence:** 2

**Summary:**

This submission introduces a probabilistic latent-variable model that predicts the full distribution of mouse V1 calcium responses using a video-encoding core and an inference model capturing a low-dimensional, stimulus-independent brain state. Compared with video-only baselines, the model (i) raises test log-likelihood, (ii) generalises better to OOD inputs when conditioned on partial population activity; (iii) the learned latents correlate with behaviour and reveal a cortical topography without supervision.

**Questions:**

1. Have you tried replacing the 3D CNN core with a recent ViT-based core such as V1T? Even a small-scale experiment would show whether the latent state provides performance gains, which would be quite important, in my opinion.

2. Could you please add standard errors or statistical significance results where they are missing?

3. On the significance of the paper: could the authors please clarify whether the inference of latent variables is performed conditioned on the neuronal responses given the same stimuli and from the same timesteps? If so, I struggle to justify the significance, or rather the claimed improvement, in the model's ability to better predict the responses of other neurons. If one assumes that a subset of neurons at timestep $t$ can inform the prediction of (or can contain information about) other neurons, then it is only natural that a model that conditions its prediction on a subset of neurons can theoretically perform better than a model that does not. As such: Is a comparison to models that do not condition on the subsets of neurons justified / fair / meaningful? I would be happy to hear your clarifications on this matter.

**Ethical Concerns:**

["NO or VERY MINOR ethics concerns only"]

**Final Justification:**

The authors have addressed my questions with new experiments and further clarifications.

I am keeping my score at borderline accept, as I do not believe I have the necessary knowledge of the related work to be confident in my assessment.

Nevertheless, having spent some time on this paper, I do not have any reason to consider rejection.

**Limitations:**

Yes

**Quality:**

3

**Strengths And Weaknesses:**

1. Quality:
- (+) The paper is technically sound with a well-formulated probabilistic model.
- (-) It would be important to include errors in Table 4. The claims cannot be fully substantiated without them.

2. Clarity:
- (+) Clearly written and well-organised
- (-) Minor typos (e.g. “non-condititoned”, “percise”).
- (-) Readers unfamiliar with ZIG would definitely benefit from a short paragraph before Eq. 1.

3. Significance:
- (+) First probabilistic model that jointly handles natural-movie input and models latent brain state for neuronal response prediction, filling a gap in dynamic encoding models;
- (+) The latent factors capture behaviour and cortical topography without supervision;
- (-) Comparisons stop at the Sensorium CNN/GRU baseline; the authors acknowledge more recent transformer cores [3, 27] but do not test on them, so the incremental benefit on modern architectures is unknown.
- (-) Impact seems questionable due to the modest gain in correlation against baselines and due to the fact that conditioning requires simultaneous recording of many neurons. This would limit immediate applicability to smaller-scale datasets.

4. Originality:
- (+) Novel combination of (a) ZIG, (b) latent-state inference from a subset of neurons, and (c) dynamic video core. Prior latent models either ignore stimulus or are image-based; the present work unifies both for video. The discovery of topographic maps in the weights that connect latents to neurons is interesting.
- Methodologically the pieces (variational auto-encoding, ZIG, 3D CNN core) are known, but the novelty lies in their integration.

---

> ### Author Rebuttal · Authors · 2025-07-31
>
> Thank you for your positive review, referring to our paper as “technically sound with a well-formulated probabilistic model” and “clearly written and well-organised”. We also appreciate your constructive feedback.
> We believe we can address all your concerns and questions:
>
> Addressing raised weaknesses:
>
> **W1**: Error bars on Table 4:
> Thanks for pointing this out. We ran the experiments again with models initialized on 2 more seeds and reported the standard error of mean along 3 seeds with the slightly changed results of Table 4 below:
> |    | Freely learned    | Position based     | Same    |
> |-|-|-|-
> | Log-Likelihood  | -0.32 (0.05)     | -0.70 (0.04)| -0.88  (0.04)
> | Conditioned Correlation | 0.24 (0.003)    | 0.22 (0.005) | 0.18 (0.003)
>
> **W2** Introducing ZIG: Thanks for your suggestion. We would incorporate an intuitive introduction like this in the text: “In calcium imaging of neurons, the measured fluorescence signals often include many zero or near-zero values when neurons are inactive, along with positively skewed values when activity occurs. A zero-inflated gamma distribution models this by combining a point mass at zero (to capture silent periods) with a gamma distribution (to represent the continuous, positive-valued calcium activity during neural firing). This allows for more accurate modeling of the sparse and bursty nature of neural activity.”
>
> **W3** Comparisons with transformer cores: See **Q1**
>
> **W4** Limited impact:  We would like to address your point regarding limited impact. First, advances in recording techniques have made large-scale datasets increasingly common, with several open-access resources now available, including MiCRONS [1], Sensorium 2023 [2], and the Allen Brain Observatory [3]. Deep learning based predictive models, shown to be superior in predictive performance, typically rely on such datasets [4], and hence we do not see data scale as a limiting factor for our approach—especially since the data we use is publicly available. Regarding the applicability to smaller-scale datasets, we believe our work can serve as a strong pretrained initialization and could adapt to smaller-scale datasets via fine-tuning.
> Second, our primary goal was not to outperform baselines on point-estimate metrics like correlation, but to more accurately model the full neuronal response distribution (log-likelihood), while additionally capturing the influence of unobserved latent states alongside stimulus-driven activity—a well-known strong signal in large-scale populations  [5].
> Lastly, there is evidence that the underlying latent state is effectively high-dimensional [5].  Therefore, it cannot easily be estimated using behavioural variables as proxy. Other existing methods ([3, 35, 40, 55, 56] from the original manuscript) either ignored the stimuli or were not probabilistic and thus ignored the noise correlations, both limiting the extraction and interpretability of the latent state vectors.
>
> Since we show that our learned latents states correlate with behavior, this allows us to design realistic in-silico experiments, supporting in-vivo causal testing. We hence believe our model trained on a large-scale dataset has a direct experimental impact.
>
> **W4** Typos:  Thanks for catching the typos! We will thoroughly go through the manuscript again.
>
> Addressing raised questions:
>
> **Q1**:
> Below, we show that the trends are qualitatively the same - e.g., the ZIG model with the ViT core is a bit worse than Poisson due to the optimization trade-off discussed in lines 199-204,  and then the latent model actually improves the performance above the Poisson baseline.
> While we did not obtain the state-of-the-art performance using ViT during rebuttal---very likely attributable to limited hyperparameter tuning---we hope that this experiment is sufficient to show that the latent method we introduced is transferable towards other cores.
> Below, we rerun the Table 1 comparison, this time replacing the 3‑D CNN with a ViT backbone for every model.
> |    | Poisson baseline   | Video-only ZIG      | Latent ZIG    |
> |-|-|-|-
> | Correlation  | 0.17 (0.007)    | 0.155 (0.009) | 0.153  (0.007)   |
> | Log-likelihood  | -     | -1.2 (0.09) | -0.45 (0.06)   |
>
> Additionally, Conditioning on half the neurons further boosts correlation up to 0.194 (0.006), showing that encoding neuron responses works with a different core, too.
>
> **Q2**: See W1
>
> **Q3**: Yes, you are right, in the ‘Conditioned’ scenario, we give a subset of responses from the same time and movie frame as input for the model. We fully agree that it would be unfair to compare such a model to a baseline that lacks this extra information. We reported this experiment to show that our model additionally **allows** us to condition on other neurons and baselines that predict neural activity to stimuli currently cannot.
> Thus, our main experimental results in Table 1 report the performance of all models without conditioning on any response. See lines 179-180 for evaluation details of this “Non-Conditioned” scenario. In the ‘Non‑Conditioned’ scenario, we still report Pearson correlation, the field’s standard metric, and our video‑only ZIG model’s correlation remains close to the Poisson baseline, while also allowing exact log probability calculation (scoring the full-response distribution) and how including the learned latents improves the log probability.
>
>
> Regarding existing works that allow conditioning on other neurons, to the best of our knowledge, there are currently no public baselines that support this. The closest is Neuroformer [6], but its original implementation does not support multisession training. Scaling it to a dataset like Sensorium 2023 would require substantial modifications—such as adjusting context window sizes and architectural changes—making it unsuitable as an out-of-the-box baseline.
>
> However, as discussed in W4, our primary goal and the significance of the paper is not to outperform baselines on correlations but to create an accurate and biologically plausible model for the entire neuronal response distribution (log-likelihood) and the latent state. Such models could enable more previously unavailable analyses, such as the cortical topography analysis from Figure 5.
>
> We hope these comments shed some light on our motivation, technical, and design choices. We also hope that it will help the reviewer reconsider some criticism.
>
> References:
> [1] Zhou, P., Reimer, J., Zhou, D., Pasarkar, A., Kinsella, I., Froudarakis, E., Yatsenko, D. V., Fahey, P. G., Bodor, A., Buchanan, J., Bumbarger, D., Mahalingam, G., Torres, R., Dorkenwald, S., Ih, D., Lee, K., Lu, R., Macrina, T., Wu, J., da Costa, N., Reid, R. C., Tolias, A. S., & Paninski, L. (2020, March 25). EASE: EM‑assisted source extraction from calcium imaging data. bioRxiv. https://doi.org/10.1101/2020.03.25.007468
>
> [2]  Turishcheva, P., Fahey, P. G., Hansel, L., Froebe, R., Ponder, K., Vystrčilová, M., Willeke, K. F., Bashiri, M., Willeke, K. F., Bashiri, M. P., Willeke, K. F., Ecker, A. S., Sinz, F. H., Ding, Z., Tolias, A. S., & Ecker, A. S. (2024, July 12). The Dynamic Sensorium competition for predicting large‑scale mouse visual cortex activity from videos. arXiv. https://doi.org/10.48550/arXiv.2305.19654
>
> [3] de Vries, S. E. J., Lecoq, J. A., Buice, M. A., Groblewski, P. A., Ocker, G. K., Oliver, M., … Koch, C. (2020). A large‑scale standardized physiological survey reveals functional organization of the mouse visual cortex. Nature Neuroscience, 23(1), 138–151. https://doi.org/10.1038/s41593-019-0550-5
>
> [4] Lurz, K.‑K., Bashiri, M., Willeke, K. F., Jagadish, A., Wang, E., Walker, E. Y., Cadena, S. A., Muhammad, T., Cobos, E., Tolias, A. S., Ecker, A. S., & Sinz, F. H. (2021, January 12). Generalization in data‑driven models of primary visual cortex, ICLR 2021
>
> [5] Stringer, C., Pachitariu, M., Steinmetz, N., Reddy, C. B., Carandini, M., & Harris, K. D. (2019). Spontaneous behaviors drive multidimensional, brain‑wide population activity. Science, 364(6437), 255. https://doi.org/10.1126/science.aav7893
>
> [6] Antoniades, A., Yu, Y., Canzano, J., Wang, W., & Smith, S. L. (2023). Neuroformer: Multimodal and multitask generative pretraining for brain data. arXiv preprint arXiv:2311.00136.

---

> > ### Comment · Reviewer_WMJZ · 2025-08-03
> >
> > Thank you for your clarifications. No more questions from me at this point.

---

### Official Review · Reviewer_qQZJ · 2025-07-03

**Clarity:** 2
**Significance:** 2
**Originality:** 3
**Rating:** 4
**Confidence:** 2

**Summary:**

The paper presents a probabilistic latent-variable framework that models neural data to videos by combining stimulus-driven features with a low-dimensional, stimulus-independent brain state, addressing variability that cannot be explained by the visual input alone. Building on a 3D CNN core and a Gaussian readout, the model predicts a Zero-Inflated-Gamma (ZIG) response distributions whose parameters are jointly modulated by the video and the inferred latent variables. Compared with Poisson or video-only ZIG baselines, it improves log-likelihood and single-trial correlation. These results demonstrate that unsupervised inference of latent variables can enhance predictive accuracy and discover biologically meaningful structure without requiring behavioural supervision.

**Questions:**

1. This paper compared a video-only ZIG-model without latent to disentangle the impact of the ZIG. Could you provide a latent-Poisson baseline using the same video core? One suggestion is to keep the CNN core and readout but predict Poisson rates, then compare likelihood and correlation with the latent-ZIG model. This will help us understand whether the observed performance gains come from the ZIG likelihood itself or from the latent design.

**Ethical Concerns:**

["NO or VERY MINOR ethics concerns only"]

**Final Justification:**

The authors' rebuttal adequately addresses my concerns.

**Limitations:**

The authors have discussed the limitations in Section 5.

**Quality:**

3

**Strengths And Weaknesses:**

**Strengths**:

1. The authors proposed a 3D video encoder with a Zero-Inflated-Gamma (ZIG) likelihood whose parameters are jointly shaped by stimulus features and a latent state. This combination goes beyond the common Poisson-mean setting, making the model capture both firing-rate modulations and trial-to-trial variability in a mathematical way.

**Weaknesses**:

1. A minor concern is that this paper spends several dense paragraphs summarizing prior methods, where many only loosely connected to the proposed latent-ZIG model, which distracts from the main ideas and makes the paper feel unfocused.  A brief overview in the main text with further details moved to the supplementary material would improve clarity and readability.

---

> ### Author Rebuttal · Authors · 2025-07-31
>
> We appreciate your review and constructive feedback. We believe we can address the raised weakness and question effectively. Specifically:
>
> Addressing raised weakness:
>
> **Clarity of related works section**: In the final manuscript, we will refine the related works section to more clearly position our contributions within the broader literature. Briefly, our method builds on core-readout models to predict dynamic neural activity to video stimuli, but introduces a probabilistic formulation that also includes learning of latent factors underlying neural activity. Existing probabilistic or latent-state models typically ignore the stimulus altogether, while stimulus-driven models are either deterministic or overlook latent states. We aim to clarify this landscape and clearly situate our work at the intersection of these approaches.
>
>
> Addressing raised questions:
>
> **Q1**: Thanks for raising this point. In fact, we had the same question. Therefore, this experiment is already in Appendix G.
> We keep the 3D CNN and Gaussian read‑out unchanged and train the latent model with a Poisson loss. As we cannot directly compute likelihood on a discrete Poisson distribution for continuous signals, we insert the means of the predicted ZIG distributions into the Poisson loss. For each neuron $i$ and timepoint $t$, our ZIG model predicts the ZIG-parameters $q_{it}(\mathbf{x};\psi)$ and $\theta_{it}(\mathbf{x};\psi)$ (as in Figure 2A) based on the video input $\mathbf{x}$ and its parameters $\psi$. With these, we analytically compute the mean of the ZIG distribution $\hat{r_{it}}\bigl((q_{it}(\mathbf{x};\psi),\theta_{it}(\mathbf{x};\psi)\bigr)$. During training, we then use Poisson Loss between those predicted responses $\hat{r_{it}}$ and the observed neural responses $r_{it}$ as an objective. Basically, we parametrize a mean firing rate (needed for Poisson) via a ZIG distribution.
> In addition to Appendix G below, we report both correlation and likelihood, showing that a latent model trained with Poisson loss has a worse likelihood performance and a similar correlation performance compared to the ZIG latent model. Moreover, adding a latent improves the likelihood compared to the video-only baselines, regardless of the used loss function.
> |    | Video-only ZIG    | Video-only Poisson      | Latent ZIG    | Latent Poisson     |
> |-|-|-|-|-
> | Correlation  | 0.183     | 0.195 | 0.183     | 0.184   |
> | Log-likelihood  | -0.98     | -1.61 | -0.30    | -1.39   |
>
>
> We hope that these comments helped to understand our paper a bit better, and we will improve the writing based on your feedback.

---

> > ### Comment · Reviewer_qQZJ · 2025-08-05
> >
> > I thank the authors for the clarifications. I have no further questions at this time.

---

### Official Review · Reviewer_FxVu · 2025-07-03

**Clarity:** 3
**Significance:** 3
**Originality:** 3
**Rating:** 4
**Confidence:** 3

**Summary:**

This paper presents a probabilistic model for predicting neural activity in the mouse primary visual cortex (V1) in response to video stimuli. The model aims to account for both the stimulus and unobserved internal brain states by incorporating a latent variable inferred from a portion of the recorded neural population. The authors evaluate this model on a public dataset, reporting improvements in log-likelihood over a video-only model. The study also analyzes the learned latent variables, showing they correlate with the mouse's behavior and have a spatial organization related to the cortical layout of the neurons.

**Questions:**

1. The justification for using a Zero-Inflated Gamma (ZIG) distribution over a Poisson one is clear, given that ZIG is better suited for the continuous and sparse nature of calcium imaging data. However, the choice of the Gamma component itself is worth probing. Have you considered other heavy-tailed distributions to model the positive responses? Do you have any intuition on whether this might help close the approximation gap?

2. Here is a suggestion for a baseline that could further contextualize your model's performance. Instead of a direct ZIG observation model, an alternative would be a more biophysically-inspired model. One could first predict underlying discrete “spikes” with a Poisson model and then apply a deterministic temporal convolution to model the slow dynamics of the calcium transients. A comparison against such a benchmark would be very interesting to see and would significantly strengthen the paper's claims about the effectiveness of your end-to-end framework.

**Ethical Concerns:**

["NO or VERY MINOR ethics concerns only"]

**Final Justification:**

The paper introduces a relatively novel probabilistic latent variable model for predicting neural activity. It is technically sound, and the experiments support the claims.

**Limitations:**

Yes

**Quality:**

3

**Strengths And Weaknesses:**

## Strengths:

 **A clear contribution:**

The paper addresses the important problem of modeling stimulus-independent variability in neuroscience. The proposed model, which combines a dynamic video encoder with a latent variable framework for predicting full response distributions, is a logical step forward in the field.

**Sufficient experimental validation:**

The authors perform a number of experiments to validate their approach. They compare against reasonable baselines, test on out-of-distribution stimuli, and perform control analyses to investigate the properties of the learned latent variables.


**Biologically relevant findings:**

A key result is that the latent variables, learned without supervision from behavior or location data, correlate with external variables like mouse behavior and cortical topography. This suggests the model is capturing meaningful structure in the neural data.


## Weaknesses:

**1. Performance on key metrics:**

A notable weakness is that the proposed model performs worse on the single-trial correlation metric compared to a simpler Poisson baseline. While the authors report an improvement in log-likelihood, the trade-off with the correlation metric, which is a standard measure of predictive accuracy in this domain, is a significant concern.

**2. Model and training complexity:**

The latent variable model introduces considerable complexity over the baselines. The appendices show that the best performance is achieved only when initializing from a pre-trained model, which suggests that training from scratch is challenging and may lead to poor solutions. This could be a practical barrier to adoption.

**3. Purity of latent states:**

The central claim is that the latents capture stimulus-independent variability. However, the authors' own analysis shows that for one of the five mice, the latents contain residual stimulus information. This finding, while reported transparently, somewhat undermines the model's core premise.

---

> ### Author Rebuttal · Authors · 2025-07-31
>
> Thank you for your review, and for viewing our work to have “clear contribution” and  “sufficient experimental validation” and providing valuable feedback.
>
> Here we address the weaknesses and questions raised during the review.
>
> Addressing raised weaknesses:
>
> **Performance on key metrics**: We acknowledge that the ZIG model slightly underperforms the Poisson baseline in lines 240-245 of our main text (L240--245) due to the optimization trade-off, since ZIG is not an exponential family distribution (see [1]), and maximizing its likelihood does not automatically maximize its correlation since mean is not a sufficient statistic of the ZIG distribution [1].
> However, our primary goal was not to outperform baselines on point-estimate predictions (correlation) but to more accurately model the entire neuronal response distribution (log-likelihood) along with latent factors, which are known to be a strong signal in neuronal populations [2]. Models that are solely based on sensory input cannot capture these noise correlations.
> We chose ZIG as a more biologically fitting distribution for deconvolved calcium trace signals, which often include many zero or near-zero values when neurons are inactive, along with positively non-integer skewed values when activity occurs, and our choice also agrees with the literature [3, 4]. Compared to Poisson, which is a discrete distribution, for ZIG, we can also compute the exact likelihood. We chose ZIG for these reasons.
> While we report the correlation to facilitate comparison with other models, we want to highlight that likelihood is a complete metric as it captures the full stimulus-conditioned distribution rather than correlation, which just captures the conditional mean. Thus, we believe likelihood aligns better with our aim to model both variability and higher moments of neural activity.
>
>
>
> **Model complexity**: We agree that our approach involves a two step procedure—training a standard video-to-neuron model before training the latent module--instead of training everything from scratch, as we show that we get better results this way (reported in Appendix L). However, we do not think that this is necessarily a weakness since using a pretrained backbone is now standard practice in many modern machine learning pipelines, models consisting of multiple separate components are seldom trained fully from scratch. Additionally, we view our two step approach to also provide us with modularity: one can simply use an existing pre-trained video-neuron encoder and fine-tune with the latent module on top, further promoting practical adoption.
>
> **Purity of latent states**: Our claim about the latent separation stems from the independence structure of the latent variables that follows a factor analysis model (Fig. 2C), where latents are a priori independent of the stimulus – giving a bias toward stimulus independence. Our objective function (ELBO), $E_z [\log p(y \mid x, z)] - \mathrm{KL}\bigl[q(z \mid y) | \mathcal{N}(0,1)\bigr]$, encourages this via the KL term that tells $z$ to remain isotropic unless it helps predict the responses. In the analysis you are referring to, we sample from the approximate posterior $q(z \mid y)$. Due to the collider structure $x \to y \leftarrow z$, the posterior $q(z \mid y)$ can still carry stimulus information. Our control experiment checks whether these samples encode stimulus signals, and we will make this subtlety clearer.
>
> Addressing raised questions:
>
> **Q1**: Thank you for this suggestion—we had very similar thoughts and in fact ran a series of follow‑up experiments in which we replaced the Gamma component of the ZIG distribution by a more flexible normalizing‑flow-based distribution, while keeping everything else fixed (latent-state and stimulus dependence). The normalizing flow consists of a Gaussian base distribution, followed by a series of invertible transforms, $T$. To capture the peak at zero, the responses below a threshold $\rho$ are still modeled by a uniform distribution as in our original ZIG setup. For each neuron, the flow has learnable parameters. Thus, for responses $y_i>\rho$ of neuron $i$,  we apply the flow $T_i$:  $ p(T_i(y_i)\mid z,x)= \mathcal{N} \left( \mu(x, z), 1 \right) $, where $z$ denotes the latent variable, $x$ the video and $\mu(x, z)$ is the predicted mean of the Gaussian.
> The actual response likelihood follows from the change--of--variables rule
> $ p(y_i \mid z,x)= p\bigl(T_i(y_i)\mid z,x\bigr) \cdot \bigl|\det\nabla_{y_i}T_i(y_i)\bigr|. $
>
> We also tested replacing the Gaussian base distribution with a Gamma positive‑tail, and a different number of flow layers, ranging from 2-9. The flow layers alternated affine with non‑linear functions, including {log⁡,exp⁡,tanh⁡,softplus,ELU}. Our best combination, a two‑layer $\text{affine}\rightarrow \text{softplus}^{-1}$ flow with a Gaussian base distribution, could not improve upon our ZIG model (see below).
>
> Thus, we concluded that the ZIG distribution is already a pretty good fit to the neuron responses. Further, we concluded that it is tricky to close the approximation gap- the trade-off between correlation and likelihood- that arises for non-exponential distributions [1].
>
> |   | ZIG Model    | Flow Model      |
> |-|-|-
> | Conditioned Correlation | 0.23     | 0.22   |
> | Log-likelihood | -0.98    | -1.2   |
>
> We will integrate these experimental results as part of the appendix.
>
>
> **Q2**:  Thank you for the interesting suggestion. We agree that a model producing discrete spikes and followed by convolutional calcium dynamics would mimic the biological process and data collection better and is scientifically interesting. We tested this briefly in the past. However, because we could not make it work well, we abandoned this idea.
>
>
> We hope these comments make our goals and choices clearer and help address some of the concerns raised.
>
> #### References:
> [1] Konstantin-Klemens Lurz, Mohammad Bashiri, Edgar Y. Walker, and Fabian H. Sinz. (2023) Bayesian oracle for bounding information gain in neural encoding models. In The Eleventh International Conference on Learning Representations.
>
> [2] Stringer, C., Pachitariu, M., Steinmetz, N., Reddy, C. B., Carandini, M., & Harris, K. D. (2019). Spontaneous behaviors drive multidimensional, brain‑wide population activity. Science, 364(6437), 255. https://doi.org/10.1126/science.aav7893
>
> [3] Zhou, Ding, and Xue-Xin Wei. "Learning identifiable and interpretable latent models of high-dimensional neural activity using pi-VAE." Advances in neural information processing systems 33 (2020): 7234-7247.
>
> [4] Zhu, F., Grier, H. A., Tandon, R., Cai, C., Agarwal, A., Giovannucci, A., ... & Pandarinath, C. (2022). A deep learning framework for inference of single-trial neural population dynamics from calcium imaging with subframe temporal resolution. Nature neuroscience, 25(12), 1724-1734.

---

> > ### Comment · Reviewer_FxVu · 2025-08-05
> >
> > Thank you for your elaborate answers, and for conducting the normalizing flow experiments. I have no further questions for now, but will follow the discussions with other reviewers.

---

### Note · Authors · 2025-08-15

We thank the reviewers for their constructive feedback. To summarize: The reviewers found the paper technically sound and clearly written, noted that the learned latent state is biologically meaningful (behavioral correlations and cortical topography), and recognized improvements in test log-likelihood and out-of-distribution performance.

The main concerns raised were:

- **Correlation–likelihood trade-off (WMJZ, FxVu)**: how to justify the slightly lower correlation performance of our models compared to Poisson baseline.
- **Absence of a latent baseline and further interpretation (FxVu, cFTH)**: how the learned latents compare to other latent-variable approaches, and what the latents represent.
- **Sensitivity to neuron selection**: if conditioning on subsets of neurons from fixed layers, affects model performance.
- **Transferability to alternative core architectures (WMJZ)**: whether the performance gains from adding latents persist when replacing the CNN core with modern architectures such as ViT.
- **Distributional assumptions (FxVu, qQZJ)**: whether using a different distributional form could yield better fits than ZIG.
- **Reporting and clarity**: requests for error bars in Table 4, and clarifications to minor points in the text.

We performed additional analyses and clarifications to address these points; we will include them in the final manuscript.
- **For the correlation–likelihood trade-off**, we argued that likelihood—our primary metric—captures the full stimulus-conditioned distribution, while correlation captures only means; our ZIG models improve likelihood while keeping correlation comparable to Poisson.
- **To address the missing latent baseline**, we trained per-mouse CEBRA models and found that our model captures similar pupil-dilation but substantially higher treadmill-speed correlations, suggesting that our model captures more behaviorally informative states.
- **For alternative likelihoods**, we trained a normalizing-flow and found it did not surpass ZIG. An additional latent-Poisson baseline confirmed that latents improve likelihood irrespective of loss.
- **With a ViT core**, the same pattern held: ZIG slightly underperformed Poisson in correlation, but adding latents improved likelihood over the stimulus-only ViT.
- Finally, we tested **neuron-selection sensitivity**, added SEM across seeds, and corrected typos.

We thank the reviewers for their thoughtful feedback and hope that we were able to address all concerns raised.

---

### Decision · Program_Chairs · 2025-09-17

**Decision:**

Accept (poster)

**Comment:**

This paper proposes a probabilistic encoding model for mouse V1 activity that combines naturalistic video input with stimulus-independent latent factors, modeling full response distributions through a Zero-Inflated Gamma (ZIG) likelihood. The key scientific finding is that these learned latents, though inferred without supervision, align with behavioral variables such as pupil dilation and locomotion and exhibit spatial organization across cortical surface. Empirically, the model improves test log-likelihood over video-only baselines, generalizes better on out-of-distribution stimuli, and achieves comparable correlation to Poisson models while offering a richer account of trial-to-trial variability. This unification of stimulus-driven and latent state modeling fills a notable gap in dynamic encoding approaches.

The strengths of the paper are multiple: (1) a well-formulated probabilistic framework that addresses stimulus-independent variability directly, (2) biologically meaningful latent states that emerge without behavioral supervision, and (3) solid experimental validation across likelihood, correlation, OOD generalization, and interpretability analyses. Reviewers appreciated that the model captures meaningful latent structure not just statistically but also in terms of behavior and cortical topography. While some weaknesses were noted—slightly lower correlation compared to Poisson baselines, training complexity, and questions about latent purity—the authors provided thoughtful clarifications and additional experiments, including normalizing-flow likelihoods, latent-Poisson baselines, neuron-selection sensitivity, and transfer to ViT cores. These helped establish robustness and clarify trade-offs.

After rebuttal and discussion, reviewers generally agreed that the benefits of the proposed approach outweigh its limitations. The correlation–likelihood trade-off is well justified, the latent states show stronger behavioral correlations than CEBRA baselines, and added experiments confirmed the method’s flexibility across architectures and datasets. Although not all concerns are fully resolved, the consensus is that this is a technically sound and biologically meaningful contribution that advances dynamic neural modeling. I therefore recommend acceptance.